# Extracellular vesicles could be a putative posttranscriptional regulatory mechanism that shapes intracellular RNA levels in *Plasmodium falciparum*

Mwikali Kioko[1,2], Alena Pance [ID][3,4], Shaban Mwangi[1], David Goulding[3], Alison Kemp[5], Martin Rono[1,6], Lynette Isabella Ochola-Oyier[1], Pete C. Bull[1], Philip Bejon[1,7], Julian C. Rayner [ID][5] & Abdirahman I. Abdi [ID][1,6,7] ✉

*Plasmodium falciparum* secretes extracellular vesicles (*Pf*EVs) that contain parasite-derived RNA. However, the significance of the secreted RNA remains unexplored. Here, we compare secreted and intracellular RNA from asexual cultures of six *P. falciparum* lines. We find that secretion of RNA via extracellular vesicles is not only periodic throughout the asexual intraerythrocytic developmental cycle but is also highly conserved across *P. falciparum* isolates. We further demonstrate that the phases of RNA secreted via extracellular vesicles are discernibly shifted compared to those of the intracellular RNA within the secreting whole parasite. Finally, transcripts of genes with no known function during the asexual intraerythrocytic developmental cycle are enriched in *Pf*EVs compared to the whole parasite. We conclude that the secretion of extracellular vesicles could be a putative posttranscriptional RNA regulation mechanism that is part of or synergise the classic RNA decay processes to maintain intracellular RNA levels in *P. falciparum*.

*Plasmodium falciparum*, the most lethal malaria parasite, still causes hundreds of thousands of deaths and millions of cases, especially in children under five years[1–6]. The asexual intraerythrocytic developmental cycle (IDC) of the parasite is responsible for the clinical manifestations of malaria and starts when parasite forms called merozoites invade erythrocytes. Inside the erythrocytes, the parasites complete a 48-h cycle from ring-like stages to larger forms called trophozoites and then to multinucleated forms called schizonts. The intra-erythrocytic schizonts finally rupture the host cells, releasing daughter merozoites into circulation to invade new erythrocytes[7].

The sequential parasite developmental steps that make up the IDC are driven by a transcriptional cascade, with most genes peaking at a specific time point in the IDC when their products are required[8–11].

However, the genome-wide homeostatic process maintaining this periodic transcription pattern is poorly understood. The activity of the only known transcription factors in *P. falciparum*, the 27 members of the Apicomplexan AP2 (ApiAP2) protein family[8,12], and epigenetic regulators[13–15], are thought to control gene transcription during the IDC[12]. Of note, 21 of the 27 ApiAP2 genes are expressed during the IDC in a stage-specific manner suggesting that these transcription factors could be involved in the gene regulation of non-ApiAP2 genes[8,12,16].

Posttranscriptional control of gene expression also seems to operate during the IDC, as supported by studies reporting a lack of synchrony between active transcription and mRNA abundance[17,18], the loosening of the chromatin structure immediately after the invasion of erythrocytes, which frees the promoter and terminator regions of

[1]Bioscience Department, KEMRI-Wellcome Trust Research Programme, Kilifi, Kenya. [2]Open University, Milton Keynes, UK. [3]Pathogens and Microbes Programme, Wellcome Sanger Institute, Cambridge, UK. [4]School of Life and Medical Science, University of Hertfordshire, Hatfield, UK. [5]Cambridge Institute of Medical Research, University of Cambridge, Cambridge, UK. [6]Pwani University Biosciences Research Centre, Pwani University, Kilifi, Kenya. [7]Centre for Tropical Medicine and Global Health, Nuffield Department of Medicine, University of Oxford, Oxford, UK. ✉e-mail: aabdi@kemri-wellcome.org

*P. falciparum* genes from histones[16,19] and the poor binding of histones to the AT-rich *P. falciparum* DNA also render the parasite genome broadly accessible to transcription during IDC[16,20–22] in a temporally-regulated manner[16]. This transcriptional permissive state during the IDC[20,21], results in almost all the 5700 plus genes being transcribed to some level, including pseudogenes and genes whose proteins are only used by gametocytes, mosquitoes and liver stages[8,9,23]. Although relatively downregulated during the IDC[24,25], the fate of the RNA transcribed from the genes with no reported function in asexual parasite growth is largely unknown. We hypothesise that *P. falciparum* could possess mechanisms to eliminate the unused or unusable RNA post-transcriptionally. This hypothesis is supported by the fact that the parasite secretes small membranous spheres called extracellular vesicles (*Pf*EVs) laden with biomolecular cargoes such as proteins, lipids, and nucleic acids[26–31]. In other eukaryotes, EVs are involved in intercellular communication but also play a role in cell homeostasis by sequestering and evacuating unwanted, obsolete, or toxic materials from cells[32–34]. Although previous studies have provided evidence that the parasite secretes RNA via *Pf*EVs[26,27], the complete transcriptome of *Pf*EVs across the IDC has not been dissected in detail, nor has it been compared to that of the secreting whole parasites (WP) in the same culture.

Here, we compare the RNA content of *Pf*EVs obtained from in vitro cultures of asexual blood parasites to that of the WP secreting them. The data suggest that the secretion of *Pf*EVs could be a post-transcriptional mechanism employed by *P. falciparum* to regulate intracellular RNA levels during IDC.

## Results

### Secretion of RNA via *Pf*EVs is a conserved biological mechanism

Malaria transcriptomes are highly regulated, with asexual blood-stage parasites showing a characteristic cascade of gene expression throughout their 48-h IDC[9–11,35]. Therefore, we hypothesised that RNA secretion via *Pf*EVs would similarly be a tightly regulated phenomenon. To interrogate this, we collected 4-hourly samples of WP and the medium in which they had been grown (referred to as culture-conditioned medium, [CCM]) from tightly synchronised cultures (Fig. 1a and Supplementary Fig. 1) of six *P. falciparum* lines derived from five isolates: four short-term laboratory-adapted clinical isolates (named KE01, KE02, KE04 and KE06), sKE01 derived from KE01, and one long-term laboratory isolate, Dd2[36], as described in the methods. To minimise carryover, all spent media were harvested at each timepoint, and the parasite pellets were resuspended in fresh media. *Pf*EVs were purified from the CCM using nanofiltration followed by ultracentrifugation[30,37], and transmission electron microscopy revealed that the pellet obtained from this process contained vesicles with features of small extracellular vesicles (Fig. 1b, c). An antibody to glycophorin-A (GYPA) stained less than 0.5% of beads conjugated with the small *Pf*EVs, suggesting that our isolation method largely excludes the GYPA+ medium *Pf*EVs that are thought to originate from the erythrocyte membrane[38] (Fig. 1d, Supplementary Fig. 2a–d). To further validate our EVs isolation protocol, we applied the method to human plasma samples where, unlike *P. falciparum*-derived small EVs, the markers of plasma small EVs are well-defined[39]. We observed that the pellets enriched from plasma samples using our small *Pf*EV isolation protocol contained the markers of small EVs (CD63 and CD9), and as expected, these conventional markers were not expressed by small *Pf*EVs, which were the focus of our study (Fig. 1d, Supplementary Fig. 2e–g). These observations indicate that our EV isolation protocol excluded erythrocyte membrane-derived medium-sized EVs (microvesicles) and albumax-derived small EVs.

RNA extracted from these samples was analysed with a bioanalyser, revealing that *Pf*EV-RNA lacked the characteristic ribosomal RNA peaks present in WP RNA (Fig. 1e), consistent with previous reports[40,41]. RNAseq data were generated from both the WP and *Pf*EVs, and gene-level abundances were estimated using Kallisto[42]. We observed no

differences in gene body coverage between *Pf*EVs and WP (Supplementary Fig. 3a, b). To allow comparison across isolates, clonally variant gene families (*var*, *rifin* and *stevor*) were excluded from the dataset due to a high level of polymorphism between isolates[43]. When we applied principal component analysis (PCA) to our normalised datasets, we observed that both WP and *Pf*EVs were strikingly cyclic, with samples from the same time points clustering together independent of the isolate (Fig. 1f, g), suggesting global transcriptional conservation. Second, we used gene-to-gene Pearson Correlation Coefficients (PCC) to interrogate further the degree of transcriptional conservation between the parasite isolates as done by previous studies[9,35]. We compared the culture-adapted clinical isolates to Dd2 (KE01 vs Dd2, sKE01 vs Dd2, KE02 vs Dd2, KE04 vs Dd2, KE06 vs Dd2) (Fig. 1h). As expected, the RNA profiles of the WP showed high conservation between isolates (Fig. 1h top panel) with an overlap of 3047 genes (58%) showing PCC greater than 0.5, and 3874 genes (74%) showing PCCs greater than 0.5 when considering any four of the five correlation pairs (Fig. 1i). Furthermore, at PCCs greater than 0.5, we found that the timing of RNA secretion via *Pf*EVs is also quite conserved (Fig. 1h bottom panel) with an overlap of 1770 (34%) positively correlated genes between the five comparisons, and 3080 genes (59%) when including overlaps between any four of them (Fig. 1j). These findings indicate that RNA secretion via *Pf*EVs is a conserved biological mechanism in *P. falciparum*.

### RNA secretion via *Pf*EVs is phase-shifted compared to intracellular RNA

To further illuminate the benefit of RNA secretion to the parasite, we compared the temporal transcriptome trends in WP versus the trends in *Pf*EVs. First, we modelled the transcriptional profiles as sinusoids using Fourier transform to calculate parameters that describe the wave-forms which include: phase (i.e. the timing of the RNA peaks), mesor (i.e. the baseline around which RNA abundance oscillates) and delta phase and delta mesor (i.e. the differences between *Pf*EVs and WP in the RNA phases and baselines, respectively) (Fig. 2a, b). Separate phaseograms of WP and *Pf*EVs transcriptomes were created by ordering the transcripts using the estimated phase (Fig. 2c, d). Consistent with previous studies[8,9], the WP transcriptomes ordered by the gene expression phase showed a high level of periodicity (Fig. 2c; Supplementary data 1). *Pf*EVs transcriptomes also exhibited striking periodic patterns, with 5048–5179 genes showing significant rhythmicity in RNA secretion (unadjusted *p*-value < 0.05) among the isolates (Fig. 2d; Supplementary data 1).

Next, we used the profiles of transcripts that were significantly periodic in both WP and *Pf*EVs (*p*-value < 0.05) to create combined phaseograms per isolate (Fig. 2e). We found that the phases of RNA abundance in *Pf*EVs were shifted compared to those of WP with the peak of expression in WP globally corresponding to the trough of RNA secretion via *Pf*EVs and vice versa (Fig. 2e; Supplementary Fig. 4a). We confirm this phase-shift using correlation analysis which shows a remarkable anticorrelation of RNA abundance in *Pf*EVs and WP (Fig. 2f). For example, the maximum expression of ring-stage genes, such as skeleton binding protein 1 (SBP1), in WP occurs at the expected early stages, while in *Pf*EVs their abundance peaks during trophozoite stages (Fig. 2g; Supplementary Fig. 4b, c). Similarly, the merozoite genes, such as merozoite surface protein 1 (MSP1), peak in abundance in *Pf*EVs at mid-late trophozoite stages and decrease towards schizogony when the expression of the same genes in WP summits (Fig. 2g; Supplementary Fig. 4b, c). This transcriptional phase shift suggests that the secretion of RNA via *Pf*EVs is not a random process.

### RNA of genes with no known function in IDC are enriched in *Pf*EVs compared to WP

To explore whether transcripts from specific subsets of genes were enriched in *Pf*EVs, we grouped all genes into six clusters (denoted c1 to

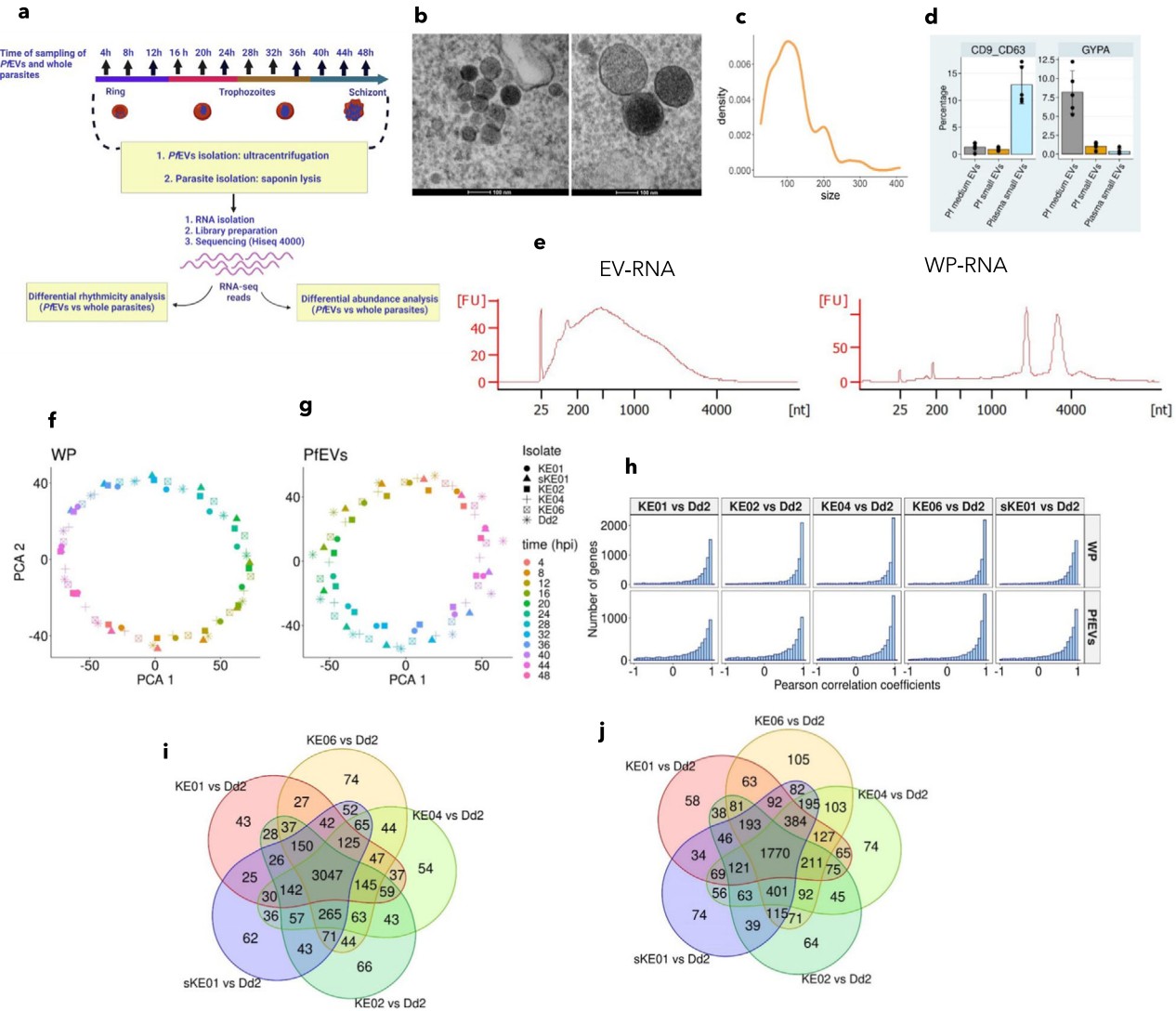

**Fig. 1 | RNA secretion via *Pf*EVs is conserved between *P. falciparum* isolates.**
**a** Schematic summary of culture-conditioned medium (CCM) sampling, sample preprocessing and data analysis (created with BioRender.com). **b** Representative transmission electron microscopy image of sectioned *P. falciparum* EVs (*Pf*EVs). Some have a conspicuously dark lumen, which shows they are rich in biological cargo. *Pf*EVs images from 10 fields were captured. **c** Density plot of *Pf*EV sizes (estimated from 177 images from 10 fields) shows that the isolated *Pf*EVs have a median diameter of 100 nm, and >90% have a diameter of <200 nm. **d** Barplots showing (left) that *Pf* small EVs, which were the focus of the current study, have relatively lower expression of classical markers of small EVs, CD9 and CD63 compared to plasma-derived small EVs. Only the proportion positive for both CD9 and CD63 was gated. Barplots showing (right) that *Pf* small EVs have lower glycophorin

(GYPA) expression than *Pf* medium EVs. The centre line represents the mean, while the limits represent ± standard deviation. The points represent *n* = 5 experiments. **e** Bioanalyser traces show that *Pf*EVs lack the ribosomal RNA peaks in whole parasites (WP). **f, g** PCA plots showing that both WP and *Pf*EVs samples assume the rhythmic circular shape of *P. falciparum* IDC transcriptomes. Samples cluster together based on the sampling time points irrespective of the isolate.
**h** Histograms showing Pearson Correlation Coefficients (PCCs) calculated from the Fourier-transformed transcriptomes between KE01, sKE01, KE02, KE04 and KE06 using Dd2 as the reference. Venn diagram analysis WP (**i**) and *Pf*EVs (**j**) gene profiles with PCC scores > 0.5 in the five comparisons of culture-adapted clinical isolates with Dd2. The high overlap implies that both WP and *Pf*EVs transcriptomes are quite conserved between the isolates.

---

c6) based on the delta mesor and consistency of enrichment across the isolates (Fig. 3a–d; Supplementary data 2). Clusters c1 (*n* = 1098) and c2 (*n* = 721) genes were enriched in *Pf*EVs in all isolates and mapped to a diverse range of biological functions primarily active during the mosquito or liver stages of the parasite, including heme[44] (e.g. PPO, UROD and ALAS) and fatty acid[45] (e.g. KASIII) synthesis, the citric acid cycle[46] (e.g. MDH and SCS-beta), formation of gametocytes[47,48] (e.g. AP2-G, GDV1), the crystalloid organelle in ookinete and young oocyst (CCp1-4), mosquito salivary gland invasion and establishment of liver stage infection and egress (e.g. CRMP1-4, CTRP, CSP, LSA-1, LSAP1 and LISP1) (Fig. 3a, d; Supplementary Fig. 5 and Supplementary data 2). In contrast, genes in clusters c5 (*n* = 911) and c6 (*n* = 1651) were preferentially retained by the WP and belonged to well-characterised biological

functions active primarily during the IDC stages of the parasites, including erythrocyte invasion[49] (e.g. RH5, MSP1, EBA175, RON5), the digestive food vacuole[50] (e.g. HRP2, HRP3, CRT, PMI-III), glycolysis[51] (e.g. ENO, GAPDH, HK), and protein processing and export to the host cell membrane[52] (e.g. KAHRP, MAHRP1, SBP1, PTP1) (Fig. 3a, d; Supplementary Fig. 5 and Supplementary data 2). Clusters c3 and c4 represented 419 and 434 genes, respectively, whose delta change in RNA abundance between *Pf*EVs and WP was inconsistent among the isolates (Fig. 3a; Supplementary data 2).

To validate further that RNAs not primarily required by the asexual blood parasites were preferentially enriched in *Pf*EVs, we classified 4279 *P. falciparum* genes as either IDC or non-IDC (gametocyte, ookinete and sporozoite) markers by reanalysing single cell

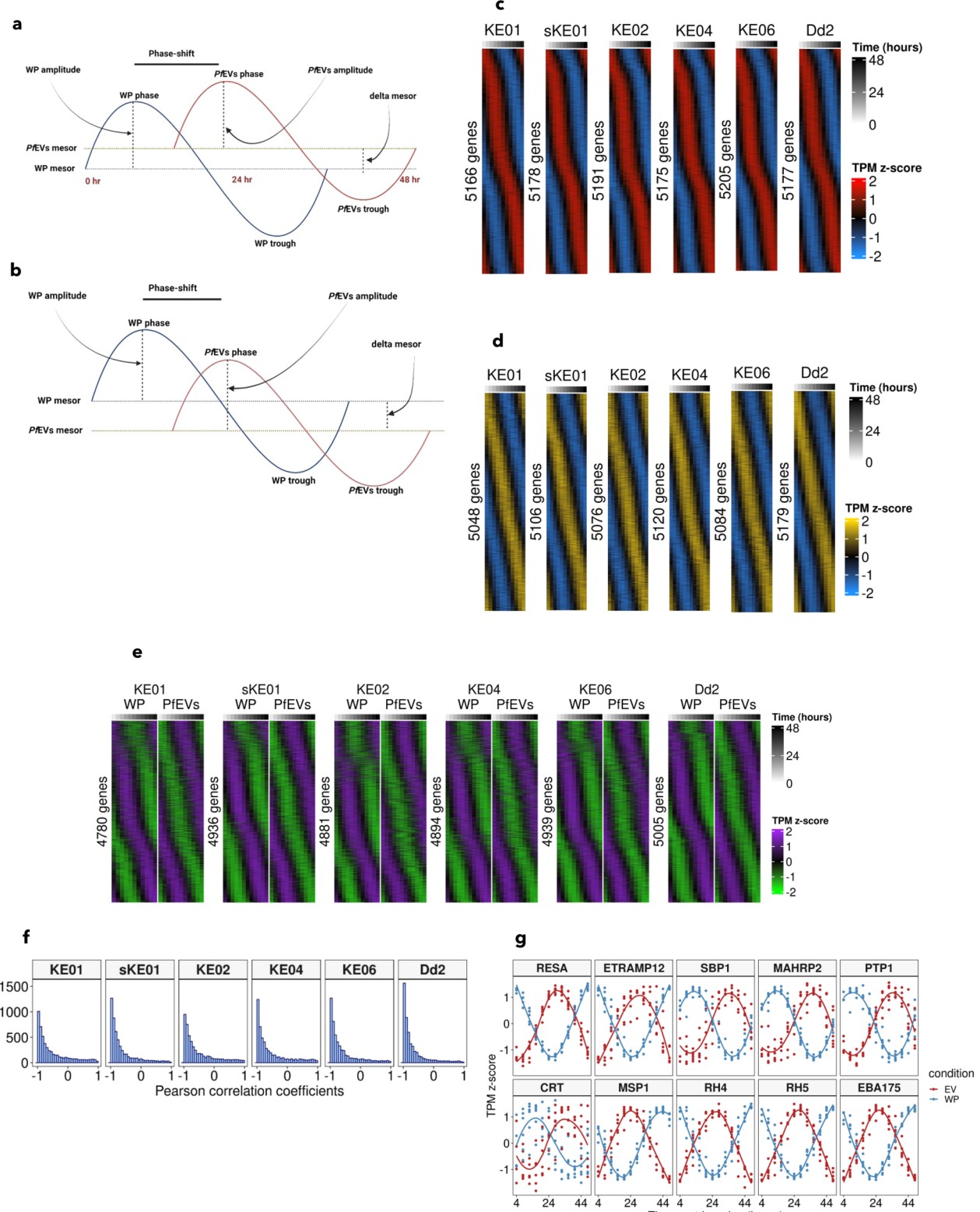

RNAseq datasets generated by the Malaria Cell Atlas studies[24,25,53]. First, we observed that clusters c1 and c2 had a strikingly higher number of genes (>85%) belonging to non-IDC stages than IDC, while clusters c5 and c6 consisted mainly of IDC-stage specific genes (~70 %) (Fig. 3e; Supplementary data 2). Second, the correlation between the log2FC (non-IDC vs IDC, Malaria Cell Atlas RNAseq) and the median delta mesor revealed that the relative gene expression in non-IDC stages was positively correlated (R = 0.44, P-value < 0.001) to the relative RNA

enrichment in PfEVs (Fig. 3f). Similarly, out of the 81 P. falciparum pseudogenes (e.g. EBA165, RH3 and ACS1b) analysed, 58 (~72%) belonged to clusters c1 and c2 while only 14 (~17%) belonged to clusters c5 and c6 (Fig. 3g).

To explore whether secretion of RNA via PfEVs could be a putative elimination mechanism of destabilised RNA in P. falciparum, we over-laid the median delta mesor (PfEVs vs WP) with RNA decay rates from a previous genome-wide mRNA dynamics study (Fig. 3h–I;

**Fig. 2 | Secretion of RNA via *Pf*EVs is rhythmic but phase-shifted compared to intracellular RNA.** Schematic representation of rhythmic parameters estimated from the RNAseq data using two scenarios (created using Biorender.com); **a** RNA abundance is higher in *Pf*EVs than WP **b** RNA abundance is lower in *Pf*EVs than WP. Phaseograms of blood-stage transcriptomes of **c** WP and **d** *Pf*EVs obtained from the *P. falciparum* isolates (KE01, sKE01, KE02, KE04, KE06 and Dd2). The phaseograms were generated from the mean-centred Fourier transformed logTPM, and the genes are ordered along the y-axis based on their phase. The periodicity of genes was tested using the likelihood ratio test, and only those that met the unadjusted *p*-value threshold of <0.05 were used to create the phaseograms. The numbers on the left of each phaseogram represent the total number of significantly (based on the likelihood ratio test) phase-shifted genes at an unadjusted *p*-value < 0.05. **e** Phaseograms show that for the vast majority of *P. falciparum* genes, the peaks of

RNA abundance in WP correspond to the trough of RNA secretion via *Pf*EVs at the global level and vice versa. Only genes detected as significantly rhythmic in both WP and *Pf*EVs were used to construct the phaseograms, and the order of genes in WP and *Pf*EVs samples is the same in each isolate. Mean z-scoring of WP and *Pf*EVs datasets was performed separately to capture the temporal trends in each compartment. **f** Histograms showing Pearson correlations between RNA abundance in *Pf*EVs and WP **g** Phase-shift demonstration using combined plots of ten well-known *P. falciparum* genes; RESA (ring infected erythrocyte surface protein), ETRAMP12 (early transcribed membrane protein 12), SBP1 (skeleton binding protein 1), MAHRP2 (membrane-associated histidine-rich protein 2), PTP1 (PfEMP1 trafficking protein 1), CRT (chloroquine resistance transporter), MSP1 (merozoite surface protein 1), RH4 (reticulocyte homologue protein 4), RH5 (reticulocyte homologue protein 5) and EBA175 (erythrocyte binding antigen 175).

Supplementary data 2)[18]. We found that RNA decay rates were positively correlated (R = 0.42, *P*-value < 0.001) to the delta mesor (Fig. 3h), with decay rates being highest and lowest for c1 and c6 genes, respectively (Fig. 3i). This implies that *Pf*EVs could be part of a posttranscriptional mechanism that controls intracellular RNA levels in *P. falciparum*. This possibility is illustrated by GDV1-antisense RNA (*PF3D7_0935390*), which inhibits the formation of sexual stages during the IDC[47] and is enriched in the WP, while its sense target, GDV1, which has no known function in asexual stages but up-regulated during gametocytogenesis[47], was enriched in *Pf*EVs (Fig. 3h). This is because the parasites were minimally exposed to stress, as the experiment was conducted over a cycle, in which the spent culture-media was replaced after every four hours with a fresh media.

Finally, we explored *var* genes expression using sKE01 line, in which two *var* genes dominate its *var* transcriptome due to selection on human brain endothelial cells (HBEC-5i). The transcripts for these two *var* genes are expected to be retained in WP by most parasites in the sKE01 culture. We mapped the RNAseq reads from sKE01 to the homologous genome (available in PlasmoDB) and found the two dominant *var* genes, *PfKE01_050037800* and *PfKE01_050038100*, to be the only *var* genes whose profile of expression in WP and secretion in *Pf*EVs is closer to that of cluster c6 genes, while the rest followed that of c1 genes (Supplementary Fig. 6). Collectively, our data indicates that a posttranscriptional process that includes secretion via *Pf*EVs could be involved in the regulation of RNA homeostasis in the *P. falciparum*.

## Discussion

*P. falciparum* secretes extracellular vesicles that contain RNA, but the significance of secreting RNA to the secreting cell is unknown. To interrogate this, we compared the RNA content of *Pf*EVs obtained from six *P. falciparum* lines to the intracellular RNA within the secreting WP. Our data suggest that *Pf*EVs play a posttranscriptional role that could be part of the parasite's mechanism to maintain the steady-state RNA levels within the parasite cells.

Our small EV-isolation protocol enriches EVs whose size range is consistent with that described for exosomes (<200 nm). Unlike the medium EVs, traditionally referred to as microvesicles, the small *Pf*EVs were devoid of the erythrocyte membrane marker, glycophorin-A, suggesting that the small *Pf*EVs may originate from inside the infected erythrocytes, including the parasite but not from the erythrocyte surface. A previous study showed glycophorin-A positivity for *Pf*EVs[54], but contrary to our study, where we filtered the CCM at 0.22 μm before ultracentrifugation, that study relied on a protocol adopted from Mbagwu et al.[55] which included CCM filtration at 0.8 μm, which does not exclude medium-sized *Pf*EVs, perhaps explaining the difference, though we cannot rule-out difference in the reactivity of the anti-glycophorin-A used. Unlike human plasma, the small *Pf*EVs did not show positivity for the classical markers of small EVs and to our knowledge, there are no validated markers for small *Pf*EVs. Future studies should therefore strive to establish protein markers that are

enriched in small *Pf*EVs compared to WP that could be used in western blotting.

Consistent with the transcriptional behaviours of malaria parasites[8–11,35], our findings reveal that transcripts from most *P. falciparum* genes are periodically secreted via *Pf*EVs during the IDC. The secretion patterns are also correlated among isolates with different backgrounds, indicating that RNA excretion via *Pf*EVs is a conserved biological mechanism that could be essential for parasite survival. Another key finding from this study was that *Pf*EVs transcriptomes are discernibly phase-shifted relative to the secreting WP, indicating that the "just in time gene expression" pattern in the WP described first by DeRisi lab[8,9] is followed by "a just in time RNA secretion" to the extracellular space. This phenomenon could be crucial in maintaining the characteristic sinusoidal patterns of RNA abundance in *P. falciparum*.

*P. falciparum* transcribes all its 5700 plus genes[8] during the asexual blood stages, including genes translated into proteins only in the sexual, mosquito and liver stages, but little is known about the destiny of RNA transcribed from non-IDC genes. Our data revealed that transcripts not known to be translated into protein during the IDC are enriched in *Pf*EVs compared to WP and have a higher decay rate than those enriched in WP. As RNA abundance in a cell is determined by the amount of transcription and the decay rate, a transcription level above the decay/elimination threshold, perhaps driven by ApiAP2 family of transcriptional factors[12], may be necessary for the genes that play critical roles during the IDC and partly explain the data in Fig. 3. Similarly, a posttranscriptional regulatory process may also play a role in determining the steady-state RNA levels[18,56,57]. The possibility of *Pf*EV involvement in regulating the parasite's intracellular RNA homeostasis is consistent with data from human umbilical vein endothelial cell (HUVEC) cultures, where RNA secretion in EVs was shown to control intracellular RNA levels[58]. Our data also aligns with the first description of EVs in the 1980s as "garbage-disposable bins" that evacuate unusable or unused biological cargo from cells[34,59–61].

Eliminating RNA via EVs could be part of or synergise the classic RNA decay mechanism to influence control of the parasite's steady-state RNA levels[62–65], as illustrated by the environmentally sensitive gene GDV1[47] and its respective regulatory transcript. However, scanty knowledge regarding the exact mechanism that loads RNA into extracellular vesicles is available. RNA loading into *Pf*EVs could be mediated by RNA-binding proteins (RBPs)[66–70]. *P. falciparum* largely relies on the posttranscriptional gene regulation mechanism centred around RBPs; it is plausible that RBPs are functionally linked to RNA loading into *Pf*EVs[67]. Consistent with this, genes encoding RBPs are expanded within the *P. falciparum* genome[71,72], perhaps to compensate for the scarcity of transcriptional factors, and in eukaryotic cells, RBPs have been implicated in the loading of RNA into EVs[73–75]. Post-transcriptional RNA modifications such as N6-methyladenosine (M6A)[76], recently reported in *P. falciparum*[65,77], may facilitate the interaction of secretion-destined RNA with RBPs[64], which could then

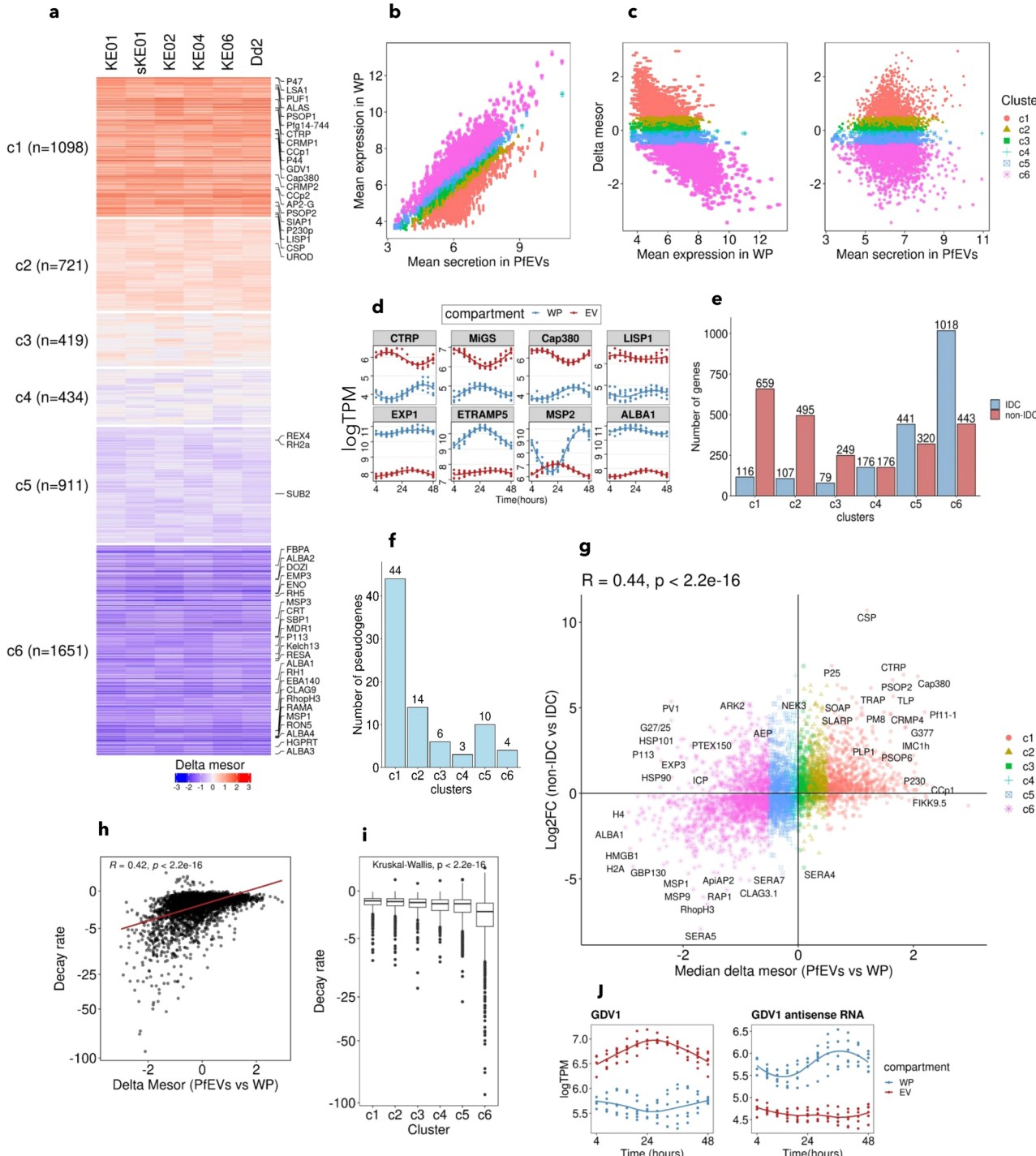

**Fig. 3 | RNA of genes with no known function in IDC is enriched in *Pf*EVs compared to WP. a** Heatmap of delta mesor obtained by comparing the RNA abundance in *Pf*EVs to that of WP. The heatmap is split into six clusters based on the median delta mesor and consistency of relative RNA abundance between *Pf*EVs and WP across isolates. Cluster c1 represents transcripts that are more abundant in *Pf*EVs than WP (median delta mesor > 0.5), while c6 represents the most excluded from secretion via *Pf*EVs (median delta mesor < −0.5). **b** A cross plot comparing mean abundance in *Pf*EVs (x-axis) to mean abundance in WP (y-axis). Each dot is a gene, and colouration is based on the clusters identified in Fig. 3a. **c** MA plots comparing delta mesor (y-axis) to mean RNA abundance in WP or *Pf*EVs. **d** Example profiles of c1 (CTRP, MiGS, Cap380 and LISP1) and c6 (EXP1,ETRAMP5,MSP2 and ALBA1) representatives. **e** Comparison of *Pf*EV-RNA relative abundance with parasite-stage specific markers obtained from the Malaria Cell Atlas data. Most genes required by non-IDC stages of the parasite are enriched in *Pf*EVs relative to

the WP, while those required during the IDC stage are enriched in the WP. **f** Most pseudogenes have higher RNA levels in *Pf*EVs than the WP (44 belong to cluster c1). **g** The relative abundance of RNA in *Pf*EVs (x-axis) positively correlates (Spearman's Rank correlation) with the relative expression in non-IDC stages (y-axis). **h** mRNA decay rates (published by Llinas lab) positively correlate (Spearman Rank correlation) with relative RNA enrichment in *Pf*EVs. **i** Boxplots showing the average decay rates in the six clusters; cluster c6 transcripts, excluded from *Pf*EVs, have the lowest decay rates. The centre lines represent the medians, limits represent the median ± interquartile range, and whiskers represent values 1.5 times above or below the 75th and 25th percentiles, respectively. Points depict values > 1.5 times and <3 times the interquartile range in each end of the boxplots. The number of genes in each cluster is the same as in Fig. 3a. **j** GDV1 antisense RNA is excluded from *Pf*EVs, while its target GDV1 is preferentially secreted.

be loaded and expelled from the parasite through EVs generated as part of non-degradative (secretory) autophagy[73] which may also exist in *P. falciparum*[78]. As the cytoplasm is the destiny of RNA for immediate translation and p-bodies for repressed RNA required for future translation[79–81], our data propose secretory EVs as the destiny of RNA to be expelled from the cell.

In summary, our data support a model in which secretion of *Pf*EVs is part of a mechanism that maintains parasite RNA homeostasis by excreting from the parasite unused (after their physiological need has been met) or unusable (not physiologically required during the IDC) RNA. Perturbation of this novel posttranscriptional RNA regulation mechanism might be detrimental to *P. falciparum*, and therefore our findings unfold new prospects to target the parasite.

## Methods

### *P. falciparum* isolates cultured

Six *P. falciparum* lines derived from five isolates were used (KE01, sKE01, KE02, KE04, KE06 and Dd2). Ethical approval was provided by the Scientific Ethics Review Unit (SERU) of the Kenya Medical Research Institute under the protocol, KEMRI/SERU/3149. Written consent was obtained from the parent/guardian of the children who donated the original clinical parasite isolates from which the short-term culture-adapted isolates used in this study were adapted. Four isolates were obtained from acute clinical malaria cases in Kenya and adapted into in vitro culture. These isolates have been in culture for less than 100 asexual cycles and were named KE01, KE02, KE04 and KE06. The sKE01 line is a product of selecting KE01 isolate to bind the HBEC-5i cell line[82] in nine passages. KE01 and KE04 were obtained from cases with impaired consciousness, while KE02 and KE06 were from uncomplicated malaria cases. The fifth was a long-term laboratory isolate called Dd2, obtained from South-East Asia[36].

### Preparation of EV-depleted parasite culture medium

Five percent (w/v) Albumax II (Gibco) was dissolved in RPMI 1640 medium and centrifuged at $150,000 \times g$ for 2 h at 4 °C (Beckmann Coulter) to deplete vesicles. The EV-depleted Albumax II was then sterilised by filtering using 0.22 μM (Millipore). The processed Albumax II was aliquoted into 50 ml portions and stored at −20 °C until use. An incomplete medium was prepared using the following: RPMI1640 (10.43 g/L), glucose (0.2%v/v), L-glutamine (2 mM), hypoxanthine (50 μg/ml), gentamicin (25 μg/ml) and HEPES (37.5 mM) (all from Gibco) under sterile conditions. To make 500 ml of incomplete medium, EV-depleted Albumax II was added to make up the complete medium. The incomplete medium was also used for the routine washing of parasite cultures.

### Culturing of *P. falciparum* asexual parasites

The asexual blood stage parasites were grown in 50 ml cultures using 75 ml culture flasks kept at 2–4% parasitaemia and 1–2% haematocrit in fresh (less than one week old) human O⁺ RBCs using the complete medium prepared above. Informed consent was obtained from the blood donors. The culture media were changed daily, and the culture flasks were gassed using a mixture of 5% O2, 5% CO2 and 90% N2. The parasite cultures were incubated at 37 °C, and the temperature of the incubators was checked twice a day. Giemsa smears were prepared every day to monitor parasitaemia. The parasites were then expanded to six 50 ml cultures in 75 ml flasks per isolate with occasional synchronisation using 5% sorbitol which kills the mature stages of the parasites.

### Harvesting of culture-conditioned medium (CCM) for *Pf*EVs isolation

Once each of the six flasks of parasite cultures reached 5–8% parasitemia at 1.5% haematocrit, they were synchronised using Percoll[83]. In brief, parasite cultures having late trophozoites (35–40 hpi) were overlaid on a 65% Percoll and centrifuged at $2500 \times g$ for 10 min with low brake. The supernatant and the RBC pellet (having mostly ring stages) were discarded while the middle layer having the mature parasites, was washed twice in the incomplete medium at $800 \times g$ for 5 min at room temperature. Giemsa smearing was performed to confirm that the asexual stage of the parasites was late trophozoites. The cultures were resuspended in fresh media while adjusting the parasitaemia and haematocrit to 4% and 1.5%, respectively. The flasks were gassed and incubated, and Giemsa smearing was performed after 15 h. If the majority (>80%) of the parasites were at the early ring stage, synchronisation using 5% D-sorbitol was performed to kill any mature parasites yet to progress to the new cycle. The parasite cultures were resuspended in complete media, gassed, and incubated at 37 °C. Four-hourly samples of culture-conditioned medium (CCM) containing parasite-derived EVs were collected into 50 ml Falcon tubes (totalling 300 ml per timepoint) and stored temporarily at 4 °C. To minimise carryover, all spent media were harvested at each timepoint, and the parasite pellets were resuspended in fresh media. A whole parasite (WP) pellet sample was also collected from each flask at each time point, digested using 0.02% saponin to exclude the red cell material, and resuspended in RNA lysis buffer. The extracted whole parasite pellets were kept at −80 °C until use, while the rest of the culture was resuspended in fresh medium, gassed, and returned to incubation at 37 °C.

### Enrichment of *Pf*EVs from CCM

Processing of CCM and isolation of *Pf*EVs was performed using an established protocol in our laboratory[30,37]. Succinctly, the collected CCM samples were centrifuged twice at $2000 \times g$ for 10 min at 4 °C to remove any remaining cells and large culture debris. The supernatant was transferred to a new 50 ml Falcon tube and centrifuged at $3600 \times g$ for 30 min, 4 °C, with the pellet again discarded. Final centrifugation at $15,000 \times g$ for 20 min at 4 °C was performed to pellet out medium *Pf*EVs (formerly called microvesicles). The debris-free CCM was passed through 0.22 μM (Millipore) to remove particles larger than 220 nm and stored at −80 °C until use.

### *Pf*EVs isolation from processed CCM

150 ml of the processed CCM was thawed, transferred to 15 ml Millipore concentration columns (50 kDa cut-off) and centrifuged at $3200 \times g$ to decrease the volume by 30 times. The concentrate was transferred to 13.5 ml Quick seal ultracentrifuge tubes (Beckmann) and centrifuged at $150,000 \times g$, for 2 h at 4 °C without brakes using a preparative ultracentrifuge (Beckmann Coulter, 70 Ti rotor). The supernatant was discarded, and pellets were resuspended in 500 μL of pre-filtered PBS and treated with 10 μL of RNase A for 15 min at room temperature. The treated pellets were washed once with 13 ml of PBS by centrifuging for 2 h at $150,000 \times g$, 4 °C and the supernatant discarded. The *Pf*EVs pellets were resuspended in RNA lysis buffer for RNA extraction and kept at −80 °C until use.

### Electron microscopy

Transmission electron microscopy was used to confirm that pellets obtained using the described method contained extracellular vesicles. Briefly, *Pf*EVs pellets were fixed in suspension in 2% glutaraldehyde and 1% paraformaldehyde in cacodylate buffer for 30 min and hard pelleted at $4000 \times g$ for 4 min, rinsed in buffer and fixed in 1% osmium tetroxide for an hour without disturbing the pellet. After rinsing again, the pellet was dehydrated in an ethanol series, staining *en bloc* with 1% uranyl acetate (at the 30% stage) and embedded in epon resin. 60 nm ultrathin sections were contrasted with uranyl acetate and lead citrate and imaged with a Tietz CCD on an FEI 120 kV Spirit Biotwin transmission electron microscope. The sizes of 177 *Pf*EVs were estimated from images of 10 fields and presented as a histogram.

## Extracellular vesicles bead-assisted flow cytometry

Qualitative analysis of small *Pf*EVs, medium *Pf*EVs (microvesicles) and plasma-derived small EVs was done using bead-assisted flow cytometry as described previously[84]. 50 µL of EVs in PBS were mixed with 1 µL of aldehyde/sulfate-latex beads (Invitrogen). 1 ml of PBS was added, and the mixture was incubated for 12 h at room temperature on a rotary wheel. Blocking was done by adding 110 µL of 1 M glycine, followed by incubation at room temperature for 30 min. The beads were pelleted by centrifugation at 2000 × *g* for 5 min, and the supernatant was removed. The pellet was washed once in 1 ml of PBS, resuspended in PBS supplemented with 0.5% foetal bovine serum (PBS + 0.5%FBS), and stained using 1× anti-CD9-APC (Cat No 341648; BD, Bioscience), 1× anti-CD63-PE (Cat No 55705; BD Bioscience) and 0.5 µL of BRIC 256 anti-235a/GYPA -FITC (Cat No 9415FI; IBGRL Research Products) for 30 min at 4 °C. Beads included with unstained EV sample, with antibodies only or with isotype antibodies were used as negative controls. The isotype control antibodies included PE mouse IgG1 (Cat No 556650, BD Bioscience) and APC mouse IgG1 (Cat No 550854; BD Bioscience). Washing was done twice with 500 µL PBS + 0.5%FBS, and the stained beads were pelleted by centrifugation at 2000 × *g* for 10 min. Data acquisition was performed using Fortessa flow cytometry (BD, Bioscience) and visualised using scatter plots (Supplementary Fig. 2a).

## RNA isolation from *Pf*EVs and whole parasites

The *Pf*EVs and WP stored in lysis buffer were thawed on ice. RNA was isolated using the Isolate II RNA Min Kit (Bioline) following the manufacturer's instructions. Once on the binding column, the RNA was treated with DNAse I to digest genomic DNA. The RNA was washed and eluted in 14 µL of nuclease-free water. The nature of extracted total RNA was assessed using Bioanalyzer Pico RNA chips (source). Messenger RNA (mRNA) was enriched from the isolated whole parasite total RNA using NEBNext® Poly(A) mRNA Magnetic Isolation Module (New England Biolabs) while following the manufacturer's instructions.

## Preparations of cDNA libraries for sequencing

cDNA libraries were prepared using the dUTP protocol[85]. Synthesis of first strand cDNA was done using Superscript III reverse transcriptase (Invitrogen) with the following cycling parameters: 25 °C for 10 min, 42 °C for 60 min and a final hold of 4 °C. Random hexamers and oligod (T) from Qiagen were used to prime first strand cDNA synthesis while dithiothreitol (DTT) was included in the reaction to stabilise the reverse transcriptase. Actinomycin D and RNase inhibitor were added to reduce antisense artefacts and prevent RNA digestion. The first strand cDNA reaction was cleaned using 1.8 volumes of RNAcleanXP beads solution and washed twice in 85 % ethanol while on the magnetic stand (Invitrogen). First-strand cDNA was eluted in 20.5 µL of Qiagen elution buffer (10 mM Tris-HCl pH 8.0), but the beads were left in the wells to be used when cleaning the second strand cDNA reaction. Unless stated otherwise, the Qiagen elution buffer was used to elute cDNA in all downstream steps.

The second strand was synthesised using NEBNext® RNA Second Strand Synthesis Module (New England Biolabs) (parameters: 16 °C for 2.5 h and a final hold of 4 °C). In the second strand synthesis reaction dUTP rather than dTTP was used. To clean the second strand cDNA, the binding properties of the beads already in the reaction were rejuvenated by adding 1.8 volumes of 20% polyethylene glycol (PEG) and 2.5 M NaCl solution. The beads were washed twice in 85% ethanol while still on the magnetic stand, and the cDNA was eluted using Qiagen elution buffer.

cDNA libraries were prepared using the NEBNext® Ultra II DNA Library Prep Kit (New England, Biolabs) following the manufacturer's instructions. First, double-stranded cDNA was enzymatically shredded to achieve an average length of 500 bp and then ligated to NEXTflex adaptor oligos with barcodes. The cDNA libraries were cleaned using 0.8 volumes of RNAcleanXP beads solution followed by two successive washes in 85% ethanol while still in the magnet. The cDNA libraries were treated with Uracil-Specific Excision Reagent (USER enzymes, Biolabs) at 37 °C for 15 min, followed by a DNA denaturation step at 95 °C for 10 min to digest the second strand. The cDNA libraries were amplified in 19 cycles using KAPA HiFi HotStart ReadyMix PCR kit (KAPABIOSYSTEMS) and P5 and P7 Illumina primers to increase yield. The PCR products were cleaned using 0.8 volumes of RNAcleanXP beads solution described above and quantified using qRT-PCR. cDNA library samples were polled in equimolar batches of 24 samples each and sequenced using the Illumina Hiseq 4000 genome analyser at the Wellcome Sanger Institute.

## RNA sequencing read quantification

Fastq files were downloaded from the sequencing pipeline, and quality parameters such as Phred scores, percentage GC content and presence of overrepresented sequences were assessed using FastQC. Assembling and summation of gene expression were performed using a super-fast aligner called *Kallisto* using default settings[42]. The *P. falciparum* 3D7 transcriptome fasta file was downloaded from PlasmoDB and used as reference[43]. There is a consensus that the expression hypervariable genes such as *var*, *rif* and *stevor* cannot accurately be compared across *P. falciparum* isolates due to a high degree of polymorphism; therefore, these were filtered from the dataset. In addition, because mRNA was enriched from the whole parasite RNA described above, all non-polyadenylated genes were excluded from further analysis. Gene body coverage was estimated using the *RSeQC* toolkit[86].

## RNAseq data normalisation

The normalisation of the RNAseq data prior to modelling in downstream steps was done to minimise systematic technical variation. First, a single data matrix containing the read counts was formed by merging the expression data from the individual samples using the R tool *tximport*. The count data were then normalised for sequencing depth and gene length by converting them to transcripts per million (TPM). TPM values were transformed using *vsn* and smoothed using *loess* algorithms in R. The RUV algorithm[87] was used to adjust the data to reduce the inherent effects of asynchronous parasite isolates. After the normalisation procedure, principal component analysis and correlation were used to assess the data.

## Estimation of rhythmic parameters

The normalised expression matrix was split into small datasets containing whole parasite or *Pf*EVs data from each isolate. Then, the RNA profile of each parasite gene was modelled as a sinusoid as previously described[8,9]. The following parameters were estimated from the rhythmic model; 1) the phase, which is the timing of gene expression in whole parasites (or secretion via *Pf*EVs) within the 48 h cycle of asexual blood stage parasites; 2) the mesor, which is the rhythm adjusted mean around which the RNA abundance oscillates and 3) the amplitude which is half the distance from the peak and trough of the cosine curve of RNA abundance. Differences between WP and *Pf*EVs regarding the rhythmic parameters were also statistically estimated and tested. All the chronobiological analysis was performed using the CircaCompare tool in R[88]. Phaseograms were created using ComplexHeatmap, and genes were ordered along the y-axis based on their phase[89].

## Functional analysis

Gene ontology terms were retrieved from the Malaria Bioconductor database. Functional enrichment analysis was carried out to identify which genes were over-represented among those enriched in *Pf*EVs relative to the whole parasite. The significance level was tested using a one-tailed Fisher's exact test (FET). The Seurat R package[90] was used to assign *P. falciparum* genes as expressed by either asexual or mosquito parasites stages (gametocytes, sporozoites and ookinetes) by reanalysing the Malaria Cell Atlas data generated by the Lawnizarc team at

the Wellcome Sanger Institute[24,25,53]. This analysis generated a log2FC (non-IDC vs IDC) that was compared to the delta mesor using Spearman correlation and boxplots. In parallel, correlation and boxplots were also used to compare RNA enrichment in *Pf*EVs to RNA decay rates from a previously published *P. falciparum* study[18]. Deconvolution was performed using singular value decomposition[91].

## Reporting summary

Further information on research design is available in the Nature Portfolio Reporting Summary linked to this article.

## Data availability

The RNAseq data generated in this study have been deposited in the gene expression omnibus (GEO) database under accession code GSE241277. The Malaria Cell Atlas reused in this study is available in the European Nucleotide Archive (accession nos. ERP021229 and ERP124136).

## Code availability

All algorithms used are published and have been cited in the method. R script used to normalise and perform the rhythmic analysis is available at https://zenodo.org/badge/latestdoi/617484246.

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

## Acknowledgements

This work was supported by Wellcome Trust Awards: 209289/Z/17/Z (to AIA), 222323/Z/21/Z and 206194/Z/17/Z (to JCR) and 203077/Z/16/Z (a core Award to KEMRI-Wellcome Trust). MK studentship was supported by the Initiative to Develop African Research Leaders (IDeAL), part of DELTAS Africa Initiative [DEL-15-003]. For Open Access purposes, the author has applied a CC-BY public copyright licence to any author-accepted manuscript version arising from this submission. The funder had no role in study design, data collection and analysis, the decision to publish, or the writing of the manuscript. We thank Prof Chris Newbold (Oxford University) for reviewing the manuscript. We also thank Dr Marcus Lee for facilitating sequencing work at Wellcome Sanger Institute.

## Author contributions

Conceptualisation: M.K., A.P., P.C.B., P.B., J.C.R., A.I.A. Methodology: M.K., A.P., S.M., D.G., A.K., M.R., L.I.O.O., P.B., J.C.R., A.I.A. Investigation: M.K., A.P., S.M., D.G., A.P., M.R., P.B., J.C.R., A.I.A. Visualisation: M.K., A.P., P.C.B., P.B., J.C.R., A.I.A. Funding acquisition: P.B., J.C.R., A.I.A. Project administration: P.B., J.C.R., A.I.A. Supervision: A.P., P.B., J.C.R., A.I.A. Writing – original draft: M.K., A.P., P.C.B., P.B., J.C.R., A.I.A. Writing – review & editing: M.K., A.P., S.M., D.G., A.K., M.R., L.I.O.O., P.C.B., P.B., J.C.R., A.I.A.

## Competing interests

The authors declare no competing interests.
