## [Peer Review File · Nature Communications]

nature portfolio

Peer Review FileReviewer comments, first round

Reviewer #1 (Remarks to the Author):

In this manuscript, Kioko et al, analyzed the RNA content of extracellular vesicles secreted by plasmodium falciparum infected red blood cells. They synchronized parasite cultures and isolated EVs every 4 hours. They compared the EV RNA content with the whole parasites at the same time point. They found that the EV RNA content is shifted compared to the intracellular RNA. They conclude that the extracellular vesicles could be a post-transcriptional regulatory mechanism to eliminate unwanted RNA from the malaria parasites. The results and conclusion are interesting and the manuscript is well written and clear. However, I have the following concerns:

- The purity of the EVs is essential to draw conclusions of the paper. The authors should demonstrate for every time points that the EVs are pure. They should show images of transmission electron microscopy. In addition, they should perform western blot with the classical EV makers. Finally, they should perform nanosight or equivalent technology to determine the size distribution and purity of the EVs. The filtration with 0.2um filter might also exclude some larger EVs.
- To my opinion the authors should also demonstrate the synchronicity of the parasites. Even a small percentage of parasites contamination can have a large impact on the transcriptome. They should demonstrate the synchronicity by Giemsa and flow cytometry.

Minor comments:

- Line 211: "...as previously described." but there is no reference
- Are human red blood cell RNA (miRNAs) identified in the transcriptome, it would make sense to use the host RNA for correlation / normalization using host RNA

Reviewer #2 (Remarks to the Author):

This manuscript provides an interesting characterization of the RNAs in *P. falciparum* extracellular vesicles (EVs) compared to the cellular RNAs content. The biological function of RNAs in malarial EVs is unknown. The main observations here are that RNAs in EVs show: i) expression periodicity along the intraerythrocytic development cycle (IDC); ii) are similar between different parasite lines; iii) are phase-shifted compared to cellular RNAs; and iv) are enriched in transcripts from genes expressed at non-IDC stages.

Main observations i and ii were somehow expected and predictable from current knowledge about the *P. falciparum* transcriptome along the life cycle: each stage is associated with a specific transcriptome, therefore the RNA content in EVs is expected to be different between different stages. Likewise, life cycle stage is by far the main source of transcriptional differences, and differences associated with life cycle progression are much larger than any other source of variation (including differences between parasite lines of different genetic backgrounds).

Observation iii is certainly an interesting observation. Observation iv is also interesting, but I do not think the authors can conclude from this data that RNAs in EVs are a parasite mechanism to excrete unused or unusable RNAs, as there are several alternative possible explanations. This interpretation, which makes it to the title, abstract and discussion, is an interesting hypothesis but is not demonstrated by this descriptive dataset. Functional assays would be needed to demonstrate this hypothesis. With the evidence presented, the title and several statements in the Abstract, Results (e.g. lines 104, 142) and Discussion about the biological role of RNAs in EVs are too speculative and not justified. Unless the authors generate new data, these statements should be toned down.

The experiments are generally technically sound. The data is nicely presented and the article is concise and well written.

Specific comments.

-Fig. 3a. Focusing on the delta mesor is an appropriate way of presenting and clustering the data, but this parameter does not capture the total expression level of the genes in each cluster. A complementary analysis focused on the total expression level of the genes in each cluster (both in EVs and in parasites) should be performed. Please note that the z scores in sup. fig. 3 do not provide this information. The majority of genes with peak expression at non-IDC stages show very low transcript levels during the IDC, orders of magnitude lower than at the stages where its protein product is required. I wonder if clusters c1 and c2 reflect changes mainly in genes expressed at near background levels in the IDC, which other authors often choose to exclude from the analyses. While the fold difference between EVs and parasites appears to be consistent, the total levels for many of these transcripts may be extremely low, even in EVs.

-In a related comment, it is possible that active exclusion of some highly abundant RNAs from EVs results in a normalization artifact that increases the TPMs for genes that are not excluded. This could reflect a biological scenario very different from the mechanism proposed by the authors: rather than unused or unusable RNAs being selectively included in EVs, it is possible that specific important RNAs required during the IDC are selectively excluded from EVs.

-Also related, statements in the Introduction and Discussion (e.g., lines 26-29, 160) do not accurately reflect what is known about *P. falciparum* gene expression. Many genes encoding proteins needed only at non-IDC stages are expressed at background levels during the IDC (orders of magnitude lower than at peak expression stages, uncertain biological significance), and some are undetectable. While posttranscriptional mechanisms certainly play important roles for some genes, it is important to make clear that the expression of genes needed at non-IDC stages is severely downregulated during the IDC.

-Lines 207-19. The percoll-sorbitol synchronization method is standard, but typically sorbitol is performed a constant number of hours (e.g. 5 h) after percoll to obtain cultures of a defined age window (e.g. 0-5 hpi, and the next time points would be 4-9 hpi, 8-13 hpi, and so on). Differences between parasite lines in the amplitude of the age window after synchronization may explain some of the differences observed among them.

Minor comments

-Line 52. Please describe clearly in the Results section if the CCM collected contained only the EVs produced from the previous time point (e.g., at the 16 hpi time point, the EVs produced between 12 and 16 hpi) or the cumulative EVs produced from the beginning of the assay. From the description in the Methods (lines 225-6) I guess the former is true, but please describe it more clearly and also in the Results. This is fundamental for the interpretation of the results of these experiments.

-Based on size or other properties, could the RNA-containing EVs be classified as mainly exosomes, or as other types of EVs?

-Lines 72-9. A PCC >0 may not be a very good indicator for high correlation. Providing numbers for genes with a PCC >0.5 would be more informative.

-Fig. 1c. Are RNAs in EVs truncated? Is it possible that they represent only fragments of RNAs rather than full-length transcripts? Is it possible that some of them correspond mainly to ncRNAs overlapping with the genes to which they are annotated? (sense or antisense). Was the distribution of reads within each annotated mRNA analyzed to explore these possibilities?

-A major source of transcriptional variation among parasite isolates or clones is clonally variant expression. In a manuscript presenting the comparison of different isolates, it would be appropriate to mention this (either in the Introduction or the Discussion).

-In a related comment, excluding the large families of clonally variant genes involved in antigenic variation is appropriate, but this important exclusion should be mentioned in the Results, not only

in the Methods. Since this exclusion involves a large number of genes, it is fundamental for the interpretation of the results presented. It explains the low level of variation between isolates.

-Fig. 2d. It may be more informative to provide combined plots (or plots for one representative strain) for a larger number of genes (e.g., ten), rather than the plots for each strain for only two genes.

-Data in Fig. 2c could be quantified as in fig. 1f, analyzing the PCC between EVs and WP to determine how many genes show a strong anticorrelation or a shifted phase.

-The origin of the field isolates should be described in more detail (geographical region where they were obtained, ethical approval, etc), or a previous reference where they were described provided.

-Panel d is missing in the legend of sup. fig. 2.

Reviewer #3 (Remarks to the Author):

The manuscript titled: "Extracellular vesicles maintain RNA homeostasis in *Plasmodium falciparum*", hypothesizes that this human malaria species could possess mechanisms to post-transcriptionally eliminate unwanted RNA via extracellular vesicles (EVs). To prove this novel hypothesis, RNAseq analysis of EVs and parasites obtained from synchronous in vitro cultures of *P. falciparum* (five different wild isolates and the Dd2 strain) at a 4-h resolution during the 48h intraerythrocytic developmental cycle (IDC), was performed. Nanofiltration followed by ultracentrifugation were used as the method to isolate EVs. A series of bioinformatics approaches, including Fourier transform, were included to demonstrate that similar to the "just-in-time expression" original IDC phaseogram of *P. falciparum*,¹ the EV content also presents a "just-in-time" RNA secretion, which interestingly is shifted compared to the transcription of parasites. Collectively, results suggest an in silico predicted association of EVs and unused mRNA such as that of sporozoites, gametocytes and ookinetes; thus, indirectly supporting the hypothesis. However, no functional evidence is presented which limits the significance of this putative post-transcriptional mechanism. Moreover, there are major and minor comments that need to be addressed before this work can be considered for publication in Nature Communications.

Major comments:

1. Due to the isolation method of EVs, nanofiltration followed by UC, the RNA cargo determined by RNAseq is not exclusively associated to EVs as this technology also isolate, to a large extent, lipoproteins. Thus, specific experiments should be considered to unequivocally show this association: (i) Bead-based flow cytometry and/or western blot against exosomal, microvesicle, apoptotic bodies and lipoproteins markers will demonstrate the degree of purity of this isolation method. (ii) SBP-1 and MSP1 were used as prime examples of this mechanism. Thus, these genes can be labelled via RNA FISH during active ring and late trophozoite transcription and later shown by high resolution imaging to be secreted in EVs. (iii) Synchronous cultures in ring stages can be treated with alpha-amanitin and 4h later, using RNAseq, conclusively demonstrate that EVs no longer contain mRNA transcribed in ring stages.

2. The transcriptome is a function of the rate of synthesis and the rate of decay. Thus, mRNA degradation has also been suggested as a putative mechanism explaining the "just-in-time" expression mechanism. In fact, transcripts for proteins that function in the same pathway or process generally have similar decay rates, implying a post-transcriptional mechanism explaining this unique expression mechanism of the IDC of *P. falciparum*.² In addition, this species has several orthologues, including the core set of proteins, that composed the machinery that degrades mRNA, unfortunately also called the exosome!³ A throughout discussion of this alternative mechanism should be included.

1. In the introduction, it is stated that "almost all 5700 plus genes within the *P. falciparum* genome are transcribed during the IDC, including pseudogenes and genes whose proteins are only used by gametocytes, mosquito and liver stages 4,5,8. The fate of the RNA transcribed from these genes

which are not involved in asexual parasite growth is unknown". This latter sentence is a wrong statement as it has been demonstrated that malaria parasites evolved translational repression (TR) as a major post-transcriptional mechanism involved in gametocytogenesis.⁴ Thus, during this stage, more than 300 genes are transcribed but remain untranslated until the mosquito stages. In this work, parasites were obtained from five different isolates; thus, it is plausible that they have different percentages of gametocytes which should be mentioned and analysed in terms of transcription of orthologues from these genes in *P. falciparum*. In the absence of this information, the Dd2 strain can be induced to produce gametocytes. Transcripts from the TR machinery should remain non-secreted. This will be another functional evidence supporting this putative post-transcriptional mechanism.

2. In a recent proteomics analysis of PfEVs from ring and trophozoite stages, the presence of unwanted proteins such as the CSP and thrombospondin were detected.⁵ It thus seems counterintuitive with the hypothesis that such unwanted mRNAs are translated in the same EVs. These results could also be mentioned in the Discussion.

3. Last, the field of EVs in malaria is mostly based in material obtained from in vitro culture in quantities that will never be produced in natural infections. Moreover, EVs from in vitro culture have been shown to contain parasite DNA representing the complete genome as well as miRNAs which are putatively involved in horizontal gene transfer of drug resistance and gametocytogenesis. It is difficult to envision that with a predicted size of 30-100 nm, these nanovesicles can physically contain all of this cargo in single vesicles. Last, circulating EVs from natural infections carry signals acting as intercellular communicators. A critical ending paragraph in the field of EVs and malaria could be considered.

Minor comments

The title does not deliver as there is no functional evidence of maintenance of RNA homeostasis through this novel putative mechanism.

Reference 1 is limited to a single site in Africa and cannot be cite accurately as a general reference for falciparum malaria.

The lipid composition mentioned in reference 14 is not associated with EVs. Please remove.

Information with regard to the percentage of gametocytes in the five isolates and Dd2 should be provided. This is of importance to demonstrate the status of transcription of gametocyte-related genes under TR in these isolates.

Depletion of EVs from albumax by ultracentrifugation was not exhaustive. It has now been demonstrated that two-hours does not suffice to fully remove EVs.

Expression of hypervariable genes was excluded from this analysis and there is a reasonable explanation for the wild isolates. Yet, the genome of the Dd2 strain is available. Thus, analysis of expression of var genes during late stages should add value to this putative post-transcriptional mechanism as these genes are not expressed, excepting one, during late IDC stages.

The quality of Fig 1 (d,f) is not of high standards making difficult to fully appreciate it.

1. Bozdech, Z., et al. The transcriptome of the intraerythrocytic developmental cycle of *Plasmodium falciparum*. *PLoS Biol* 1, E5 (2003).
2. Shock, J.L., Fischer, K.F. & DeRisi, J.L. Whole-genome analysis of mRNA decay in *Plasmodium falciparum* reveals a global lengthening of mRNA half-life during the intra-erythrocytic development cycle. *Genome Biol* 8, R134 (2007).
3. Hughes, K.R., Philip, N., Starnes, G.L., Taylor, S. & Waters, A.P. From cradle to grave: RNA biology in malaria parasites. *Wiley Interdiscip Rev RNA* 1, 287-303 (2010).
4. Mair, G.R., et al. Regulation of sexual development of *Plasmodium* by translational repression.

Science 313, 667-669 (2006).

5. Vimonpatranon, S., et al. Extracellular Vesicles Derived from Early and Late Stage. J Clin Med 11(2022).

Response to Reviewers, second round -

Reviewer 1

Comment #1: *The purity of the EVs is essential to draw conclusions of the paper. The authors should demonstrate for every time points that the EVs are pure. They should show images of transmission electron microscopy. In addition, they should perform western blot with the classical EV makers. Finally, they should perform nanosight or equivalent technology to determine the size distribution and purity of the EVs. The filtration with 0.2um filter might also exclude some larger EVs.*

Reply: We recognise that particulates derived from bursting infected erythrocytes could contribute to the *PfEVs*' RNA set. This would have been observed during the schizont stage, where the risk of contamination from the bursting schizonts is highest. Our data show that genes expressed at this stage are preferentially excluded from *PfEVs*, suggesting that our EV isolation protocol minimised this source of contamination. And if there were substantial contribution from contaminants, we wouldn't have gotten the clear cut data (at least at observation level) presented in **figure 2 and 3**.

It is a near consensus that all protocols currently used for EV isolation are for enrichment, not purification. Cognisant of this, we included an RNase incubation step in our protocol (line 302-303) to enrich RNA packaged within the lumen of *PfEVs*. We would also like to note that classic markers of exosomes (e.g., CD9, CD63, CD81, TSG101) used in mammalian systems are absent in *Plasmodium falciparum*, and the field is yet to establish agreed-upon markers of the small EVs released by the parasite. As an alternative, we applied our EV isolation protocol to a plasma sample, which, unlike *PfEVs*, has a well-established surface marker of exosomes. The plasma EVs obtained contain classic exosome markers, CD9 and CD63 (**Supplementary Fig. 2a**), suggesting that our protocol can isolate exosome-like vesicles from culture-conditioned media. We have captured the above in lines 72 -76 and **Supplementary Fig 2a**) and described method (lines 317-330)

We do not think it is practicable to include electron micrographs of every sample, every time point, and every parasite (n = 70), and it is unclear how

that would confirm the purity of the EVs. This also applies to western blot for EV markers, as detecting a marker only proves that a proportion of EVs carry it but does not prove purity, especially for *P. falciparum*, where no validated protein markers enriched in EVs compared to the secreting parasites are available. We agree that a NanoSight, would have helped determine the sizes of the isolated vesicles. Unfortunately, we do not have access to this equipment. However, our protocol has been widely used and reported in the field for isolating small extracellular vesicles, demonstrating the robustness and reproducibility of this method.

We used a 220nm filter to enrich the small EVs (<200 nm), previously called exosomes. We targeted small EVs as they are thought to originate partly from secretive (non-degradative) autophagic processes(1) involved in maintaining the homeostasis of macromolecules in cells. On the other hand, the larger than 200 nm (previously called microvesicles) form from outward invagination of the plasma membrane and hence are likely to encapsulate materials randomly depending on their proximity to the cell membrane. For this reason, we aimed to enrich small extracellular vesicles because the objective of this work is to understand the shedding of parasitic vesicles and not vesicles originating from the erythrocyte membrane.

Comment #2: *To my opinion the authors should also demonstrate the synchronicity of the parasites. Even a small percentage of parasites contamination can have a large impact on the transcriptome. They should demonstrate the synchronicity by Giemsa and flow cytometry.*

Reply: As we stated in response to comment#1, the chance of contamination from the parasite is highest during the schizont stage but genes transcribed during this stage are excluded from the EVs compared to the parasite (**Fig.3**). We have added Giemsa images (**Supplementary Fig. 2**) for four-time points from each parasite isolate to show that synchronicity was achieved. We show in **Fig 2d** that the timing of expression of 10 genes is similar in all the parasite samples and consistent with the reported literature. Thus, the transcriptomes are highly synchronous, consistent with the Giemsa images.

Comment #3: Line 211: "...as previously described." but there is no reference

Reply: We have added the reference (line 268 of the new version).

Comment #4: Are human red blood cell RNA (miRNAs) identified in the transcriptome, it would make sense to use the host RNA for correlation / normalisation using host RNA

Reply: We did not sequence miRNA. Our sequencing protocol targeted RNA species larger than 150nt. We found appropriately low levels of human RNA species (>150nt) in our data, which we think could originate from RBCs and traces of WBCs left in the blood used for culturing the parasites.

Reviewer 2

Comment #1: Main observations i and ii were somehow expected and predictable from current knowledge about the *P. falciparum* transcriptome along the life cycle: each stage is associated with a specific transcriptome, therefore the RNA content in EVs is expected to be different between different stages. Likewise, life cycle stage is by far the main source of transcriptional differences, and differences associated with life cycle progression are much larger than any other source of variation (including differences between parasite lines of different genetic backgrounds).

Reply: We agree with the reviewer's observation that the secretion of RNA via EVs is expected to be periodic, but it is important to note that the data presented in this manuscript is the first to demonstrate it.

Comment #2: Observation iii is certainly an interesting observation. Observation iv is also interesting, but I do not think the authors can conclude from this data that RNAs in EVs are a parasite mechanism to excrete unused or unusable RNAs, as there are several alternative possible explanations. This interpretation, which makes it to the title, abstract and Discussion, is an interesting hypothesis but is not demonstrated by this descriptive dataset. Functional assays would be needed to demonstrate this hypothesis. With the

evidence presented, the title and several statements in the Abstract, Results (e.g. lines 104, 142) and Discussion about the biological role of RNAs in EVs are too speculative and not justified. Unless the authors generate new data, these statements should be toned down.

Reply: We agree with the reviewer that further experiments will be required to demonstrate our hypothesis functionally. We have toned down the language in the new version of the manuscript to indicate that it is a suggested mechanism derived from the observational data. We also strengthened further the data that informed the title (**Figure 2**)

Comment #3: *Fig. 3a. Focusing on the delta mesor is an appropriate way of presenting and clustering the data, but this parameter does not capture the total expression level of the genes in each cluster. A complementary analysis focused on the total expression level of the genes in each cluster (both in EVs and in parasites) should be performed. Please note that the z scores in sup. fig. 3 do not provide this information. The majority of genes with peak expression at non-IDC stages show very low transcript levels during the IDC, orders of magnitude lower than at the stages where its protein product is required. I wonder if clusters c1 and c2 reflect changes mainly in genes expressed at near background levels in the IDC, which other authors often choose to exclude from the analyses. While the fold difference between EVs and parasites appears to be consistent, the total levels for many of these transcripts may be extremely low, even in EVs.*

Reply: Delta mesor is not imputed from the z-scores. It is the logarithmic difference between the mean secretion of RNA in EVs and the mean expression in parasites in the TPM units. We have performed the requested complementary analysis using logTPM as previously done by researchers working on mammalian EVs (2). The mean secretion via PfEVs correlates with the mean expression in the WP (**Supplementary Fig 5a**). In addition, it is improbable that the entire geneset in clusters c1 and c2 represent lowly expressed genes that other researchers exclude. It is well known that the genome of *P. falciparum* is mainly in a transcriptionally permissive state during the IDC(3, 4) (we captured this in the introduction line 39-42), and

we are unaware of any previous *P. falciparum* RNAseq study that excluded such a high number (~2000) of genes from their downstream analysis.

Comment #4: *In a related comment, it is possible that active exclusion of some highly abundant RNAs from EVs results in a normalisation artifact that increases the TPMs for genes that are not excluded. This could reflect a biological scenario very different from the mechanism proposed by the authors: rather than unused or unusable RNAs being selectively included in EVs, it is possible that specific important RNAs required during the IDC are selectively excluded from EVs.*

Reply: There does not seem to be a normalisation artefact, as genes in clusters c1 and c2 do not necessarily have higher TPMs in EVs than those in c5 and c6 (**supplementary Fig. 5a** generated in response to this comment). The RNA abundance in EVs and WP is highly correlated in all clusters, which is consistent with the literature (2).

Comment #5: *Also related, statements in the Introduction and Discussion (e.g., lines 26-29, 160) do not accurately reflect what is known about *P. falciparum* gene expression. Many genes encoding proteins needed only at non-IDC stages are expressed at background levels during the IDC (orders of magnitude lower than at peak expression stages, uncertain biological significance), and some are undetectable. While posttranscriptional mechanisms certainly play important roles for some genes, it is important to make clear that the expression of genes needed at non-IDC stages is severely downregulated during the IDC.*

Reply: We concur with the reviewer that the steady-state RNA levels of non-IDC genes are, on average, lower during the asexual blood stages than IDC-specific genes but detectable. We have clarified this in the introduction (lines 43-44) and demonstrated it in supplementary **Fig. 5b**.

Comment #6: *Lines 207-19. The percoll-sorbitol synchronisation method is standard, but typically sorbitol is performed a constant number of hours (e.g. 5 h) after percoll to obtain cultures of a defined age window (e.g. 0-5 hpi,*

and the next time points would be 4-9 hpi, 8-13 hpi, and so on). Differences between parasite lines in the amplitude of the age window after synchronisation may explain some of the differences observed among them.

Reply: The transcriptomes (**Fig.1d, Fig.3d**) and Giemsa images (**supplementary Fig 1**) support the accuracy of our synchronisation approach. However, there may not be a synchronisation approach that can achieve zero variations in the time window. The cycling period is also inherently different between the isolates, with Dd2 and KE01 having a cycle period of about 44 hours while the rest have ~48 hours. Calculation of phase-shift and delta mesor is done for each parasite isolate individually; therefore, differences in age window are unlikely to affect these two parameters, which were the main focus of our study.

Comment #7: -Line 52. Please describe clearly in the Results section if the CCM collected contained only the EVs produced from the previous time point (e.g., at the 16 hpi time point, the EVs produced between 12 and 16 hpi) or the cumulative EVs produced from the beginning of the assay. From the description in the Methods (lines 225-6) I guess the former is true, but please describe it more clearly and also in the Results. This is fundamental for the interpretation of the results of these experiments.

Reply: The reviewer's guess was correct. The EVs in each timepoint are not cumulative. All culture media were harvested at each time point and replaced with fresh media. We have described this more clearly in the Results (lines 68 to 69) and methods (lines 282 to 283).

Comment #8: *Based on size or other properties, could the RNA-containing EVs be classified as mainly exosomes or as other types of EVs?*

Reply: Although our isolation protocol targeted enriching for EVs in the size range of exosomes, we cannot exclude microvesicles as their sizes overlap. Tetraspanins (e.g., CD9 and CD63) and endocytic markers (e.g., TSG101) used as conventional markers for classifying mammalian EVs as exosomes have not been found in *Plasmodium falciparum* and consensus exosome

specific parasite markers are not currently available. Therefore, we used the term extracellular vesicles instead of exosomes to describe the vesicles.

Comment #9: *Lines 72-9. A PCC >0 may not be a very good indicator for high correlation. Providing numbers for genes with a PCC >0.5 would be more informative.*

Reply: We have increased the PCC to 0.5, as suggested by the reviewer and modified **Fig.1g-h** and the relevant text accordingly (lines 92-99).

Comment #10: *Fig. 1c. Are RNAs in EVs truncated? Is it possible that they represent only fragments of RNAs rather than full-length transcripts? Is it possible that some of them correspond mainly to ncRNAs overlapping with the genes to which they are annotated? (sense or antisense). Was the distribution of reads within each annotated mRNA analysed to explore these possibilities?*

Reply: EVs may contain primarily RNA fragments rather than full-length transcripts. However, in PfEVs, PCR was used to detect the full-length mRNA of at least one gene (a member of the ETRAMP family) (5). Mature full-length mRNA and lncRNA were also shown to be present in EVs derived from a mammalian cell-line(6).

It is possible that some of the RNA in EVs are ncRNAs. However, our library preparation protocol was not strand-specific; hence, we cannot tease out sense from antisense transcription using the current data. The RNA-seq reads map on both DNA strands, making it impossible to tell whether they are sense or antisense. In future, we will explore a strand-specific protocol to investigate the above possibilities.

Concerning the distribution, we determined the gene body coverage for two isolates (KE01 and KE02) and found no difference between WP and PfEVs (**supplementary Fig 2b-c**, lines 81 to 82).

Comment #11: *A major source of transcriptional variation among parasite isolates or clones is clonally variant expression. In a manuscript presenting the comparison of different isolates, it would be appropriate to mention this*

(either in the Introduction or the Discussion). In a related comment, excluding the large families of clonally variant genes involved in antigenic variation is appropriate, but this important exclusion should be mentioned in the Results, not only in the Methods. Since this exclusion involves a large number of genes, it is fundamental for the interpretation of the results presented. It explains the low level of variation between isolates.

Reply: We have mentioned this in the results (lines 82-84).

Comment #12: Fig. 2d. It may be more informative to provide combined plots (or plots for one representative strain) for a larger number of genes (e.g., ten), rather than the plots for each strain for only two genes.

Reply: We have generated combined line plots for ten genes (**Fig 2d**)

Comment #13: Data in Fig. 2c could be quantified as in fig. 1f, analysing the PCC between EVs and WP to determine how many genes show a strong anticorrelation or a shifted phase.

Reply: This is a good suggestion. We have presented data in **Fig. 2c** using histograms similar to **Fig.1f**. The histograms (**Fig 2e**) are skewed towards negative PCCs suggesting that most genes are inversely correlated between PfEVs and whole parasites (line 118-120). We have also plotted histograms showing the phase shift (**supplementary Fig.4a**); most genes in each isolate are shifted by more than 12 hours.

Comment #14: The origin of the field isolates should be described in more detail (geographical region where they were obtained, ethical approval, etc), or a previous reference where they were described provided.

Reply: Details about the isolates have been added to the methods section (lines 238-245).

Reviewer 3

Comment #1. Due to the isolation method of EVs, nanofiltration followed by UC, the RNA cargo determined by RNAseq is not exclusively associated to EVs as this technology also isolate, to a large extent, lipoproteins. Thus,

specific experiments should be considered to unequivocally show this association: (i) Bead-based flow cytometry and/or western blot against exosomal, microvesicle, apoptotic bodies and lipoproteins markers will demonstrate the degree of purity of this isolation method. (ii) SBP-1 and MSP1 were used as prime examples of this mechanism. Thus, these genes can be labelled via RNA FISH during active ring and late trophozoite transcription and later shown by high resolution imaging to be secreted in EVs. (iii) Synchronous cultures in ring stages can be treated with alpha-amanitin and 4h later, using RNAseq, conclusively demonstrate that EVs no longer contain mRNA transcribed in ring stages.

Reply: We do not expect problematic quantities of lipoproteins in our *PfEVs* pellets because our parasites were grown in plasma-free RBCs, and *Plasmodium falciparum* has not been shown to produce lipoproteins. One could argue that traces of lipoproteins could be left after washing RBCs with RPMI, but such remnants of lipoproteins would not harbour parasite RNA. *P. falciparum* does not encode classic markers of exosomes (e.g. tetraspanins CD9 and CD63), and therefore we do not have widely accepted markers of parasite-derived EVs that we can use. Instead, we applied our protocol on plasma sample and used the bead-assisted flow cytometry to demonstrate our protocol can enrich for EVs expressing the characteristic surface markers (CD63 and CD9) (**Supplementary Figure 2a**).

We thank the reviewer for suggesting experiments (RNA-FISH and alpha amanitin) that we can use to validate our findings. While we think these experiments would strengthen our manuscript, we have opted to tone down too firm conclusions because such data cannot be provided at this time. We also note that transcriptional inhibition can also lead to compensatory changes to RNA turnover(7, 8).

Comments #2. *The transcriptome is a function of the rate of synthesis and the rate of decay. Thus, mRNA degradation has also been suggested as a putative mechanism explaining the "just-in-time" expression mechanism. In fact, transcripts for proteins that function in the same pathway or process*

generally have similar decay rates, implying a posttranscriptional mechanism explaining this unique expression mechanism of the IDC of P. falciparum. In addition, this species has several orthologues, including the core set of proteins, that composed the machinery that degrades mRNA, unfortunately also called the exosome!3 A throughout Discussion of this alternative mechanism should be included.

Reply: Our data do not rule out other established posttranscriptional mechanisms, such as RNA binding proteins (RBPs) and small regulatory RNA but provide an additional layer that could either be part of the already established mechanisms or an independent mechanism that synergises the classic decay mechanisms. We discussed this (lines 197-229). Additionally, the relative enrichment of RNA in PfEVs (delta mesor) positively correlates with mRNA decay rates (**Fig 3f**), suggesting that secretion could be a good proxy for mRNA destabilisation (159-169 and 197-229).

Comments #3. *In the introduction, it is stated that "almost all 5700 plus genes within the P. falciparum genome are transcribed during the IDC, including pseudogenes and genes whose proteins are only used by gametocytes, mosquito and liver stages 4,5,8. The fate of the RNA transcribed from these genes which are not involved in asexual parasite growth is unknown". This latter sentence is a wrong statement as it has been demonstrated that malaria parasites evolved translational repression (TR) as a major posttranscriptional mechanism involved in gametocytogenesis. Thus, during this stage, more than 300 genes are transcribed but remain untranslated until the mosquito stages*

Reply: We concur that posttranscriptional regulation in *Plasmodium falciparum* involves various mechanisms, including RNA binding proteins, ribosome association, nucleases and antisense RNA degradation. The example given by the reviewer involves 300 genes (transcribed by very late gametocytes) which are translationally repressed by RNA-binding proteins until a mosquito takes in the gametocytes (9). To our knowledge, such a mechanism of translational repression is yet to be demonstrated during the IDC. Furthermore, a previous study found that transcription and translation

are tightly coupled during the IDC (10), indicating that translational repression may not be biologically required during the IDC. Moreover, the characteristic sinusoidal patterns of *Plasmodium falciparum* transcription do not support the stabilisation of RNA encoded by non-IDC stages during the IDC, as all genes have a single peak and a single trough, regardless of whether they are not physiologically required at this stage of the parasite (11). One could argue that the transcription rate is responsible for the sinusoidal gene expression patterns in *Plasmodium falciparum*. However, a recent study observed active gene transcription throughout the parasite cycle for most *P. falciparum* genes (12). This phenomenon is thought to occur because most parasite DNA is not bound by histones during the IDC and is thus exposed to the transcriptional machinery (13).

Comment #4: *In this work, parasites were obtained from five different isolates; thus, it is plausible that they have different percentages of gametocytes which should be mentioned and analysed in terms of transcription of orthologues from these genes in P. falciparum. In the absence of this information, the Dd2 strain can be induced to produce gametocytes. Transcripts from the TR machinery should remain non-secreted. This will be another functional evidence supporting this putative posttranscriptional mechanism.*

Reply: The five isolates were not ex vivo (they were not obtained straight from the patients). Although they were adapted to lab conditions more recently than Dd2, they have been in culture for at least 100 cycles (line 240). We have estimated the relative proportions of gametocytes using deconvolution approaches and found that they are appropriately low (compared to EVs) and do not significantly differ between the parasite strains (**Supplementary Fig. 6a**). We validated our deconvolution approach using Lopez-Barragan data (14), which has two stages of gametocytes (II and V) in addition to four-time points of the IDC (**supplementary Fig 6b**). We also think that during sexual parasite stages, physiologically required transcripts, such as those of the translational repression machinery in

gametocytes, are excluded from *PfEVs*. However, we do not have - data to support this hypothesis.

Comment #5: In a recent proteomics analysis of *PfEVs* from ring and trophozoite stages, the presence of unwanted proteins such as the CSP and thrombospondin were detected.

It thus seems counterintuitive with the hypothesis that such unwanted mRNAs are translated in the same EVs. These results could also be mentioned in the Discussion.

Reply: Translation within EVs has not been reported and EVs are generally devoid of transfer RNA which is necessary for translation to happen. Therefore, we are not sure how CSP protein can find its way into EVs in an in vitro culture of blood-stage parasites. And we have not detected CSP protein in proteome data we and others have generated. We opted not to comment on the data the reviewer pointed out to us.

Comment #6: Last, the field of EVs in malaria is mostly based in material obtained from in vitro culture in quantities that will never be produced in natural infections. Moreover, EVs from in vitro culture have been shown to contain parasite DNA representing the complete genome as well as miRNAs which are putatively involved in horizontal gene transfer of drug resistance and gametocytogenesis. It is difficult to envision that with a predicted size of 30-100 nm, these nanovesicles can physically contain all of this cargo in single vesicles. Last, circulating EVs from natural infections carry signals acting as intercellular communicators. A critical ending paragraph in the field of EVs and malaria could be considered.

Reply: EVs are heterogenous in size (30-200nm) and most likely heterogeneous in content. In addition, the largest exosome (~200 nm) is almost double the size of a giant virus. Therefore, we expect *PfEVs* to have enough space to contain several molecules of mRNAs and other macromolecules, but we don't expect a single EV to contain all the RNA molecules of a cell. However, the total population of the EVs may cover all the RNA molecules of a cell but at varying degrees of quantities.

Comment #7: The title does not deliver as there is no functional evidence of maintenance of RNA homeostasis through this novel putative mechanism.

Reply: Because we do not have functional evidence of the proposed mechanism of RNA elimination, we toned down the title to reflect that it is a suggested mechanism. However, we further strengthened the data that informed the title (**Figure 2d and supplementary fig.4**)

Comment #8: Reference 1 is limited to a single site in Africa and cannot be cite accurately as a general reference for falciparum malaria.

Reply: We have added more citations of malaria epidemiology, two from additional sites in Africa, one from Latin America and one covering malaria prevalence in ten Asian countries.

Comment #9: The lipid composition mentioned in reference 14 is not associated with EVs. Please remove.

Reply: Gulati et.al; (now citation 22 in the new version) analysed the lipid content of *PfEVs* (which they termed microvesicles), although their title does not suggest that.

Comment #10: Information with regard to the percentage of gametocytes in the five isolates and Dd2 should be provided. This is of importance to demonstrate the status of transcription of gametocyte-related genes under TR in these isolates.

Reply: We have used deconvolution approaches to show that the whole parasites were mainly asexual stages, despite the *PfEVs* containing a relatively higher proportion of sexual stage-specific RNA (**Supplementary Fig 6**). This makes sense since the parasite cultures were minimally exposed to stress as they were supplied with fresh media after every four hours.

Comment #11: Depletion of EVs from albumax by ultracentrifugation was not exhaustive. It has now been demonstrated that two-hours does not suffice to fully remove EVs.

Reply: We agree there is no winner among the EV depletion or enrichment protocols. Even the best protocols are usually not exhaustive. However, the possibility of the remaining Albumax-derived EVs harbouring *Plasmodium*-derived RNAs is minimal and therefore, any traces of Albumax-derived EVs would not affect the signal we see from our data.

Comment #12: Expression of hypervariable genes was excluded from this analysis and there is a reasonable explanation for the wild isolates. Yet, the genome of the Dd2 strain is available. Thus, analysis of expression of var genes during late stages should add value to this putative posttranscriptional mechanism as these genes are not expressed, excepting one, during late IDC stages.

Reply: Indeed, the genomes of both KE01 and Dd2 are available in PlasmoDB. As mentioned in the methods section, sKE01 was produced by selecting KE01 for binding to brain endothelial cells and, therefore, can also be mapped to the KE01 genome to analyse the var genes and a higher proportion of the parasite population express the var variant(s) that mediate binding to endothelial cells. When we compared the RNA abundance of var genes in PfEVs and WP in sKE01, we found that except for two variants (*PfKE01_050037800* and *PfKE01_050038100*) upregulated in WP following selection, the rest were enriched in the PfEVs compared to the WP (**supplementary Fig.7**). This supports our earlier observation (**Fig 3a**) that unwanted transcripts are enriched in PfEVs compared to the WP.

Comment #13: The quality of Fig 1 (d,f) is not of high standards making difficult to fully appreciate it.

Reply: The quality of the figures has been improved.

References

1. A. M. Leidal *et al.*, The LC3-conjugation machinery specifies the loading of RNA-binding proteins into extracellular vesicles. *Nat Cell Biol* **22**, 187-199 (2020).
2. T. O'Grady *et al.*, Sorting and packaging of RNA into extracellular vesicles shape intracellular transcript levels. *BMC Biology* **20**, 72 (2022).
3. A. M. Salcedo-Amaya *et al.*, Dynamic histone H3 epigenome marking during the intraerythrocytic cycle of *Plasmodium falciparum*. *Proc Natl Acad Sci U S A* **106**, 9655-9660 (2009).
4. L. Michel-Todó *et al.*, Patterns of Heterochromatin Transitions Linked to Changes in the Expression of *Plasmodium falciparum* Clonally Variant Genes. *Microbiol Spectr* **11**, e0304922 (2023).
5. X. Sisquella *et al.*, Malaria parasite DNA-harboring vesicles activate cytosolic immune sensors. *Nature Communications* **8**, 1985 (2017).
6. T. O'Grady *et al.*, Sorting and packaging of RNA into extracellular vesicles shape intracellular transcript levels. *BMC Biol* **20**, 72 (2022).
7. R. Gelfand, G. Attardi, Synthesis and turnover of mitochondrial ribonucleic acid in HeLa cells: the mature ribosomal and messenger ribonucleic acid species are metabolically unstable. *Mol Cell Biol* **1**, 497-511 (1981).
8. T. Chujo *et al.*, LRPPRC/SLIRP suppresses PNPase-mediated mRNA decay and promotes polyadenylation in human mitochondria. *Nucleic Acids Res* **40**, 8033-8047 (2012).
9. G. R. Mair *et al.*, Regulation of sexual development of *Plasmodium* by translational repression. *Science* **313**, 667-669 (2006).
10. F. Caro, V. Ahyong, M. Betegon, J. L. DeRisi, Genome-wide regulatory dynamics of translation in the *Plasmodium falciparum* asexual blood stages. *Elife* **3**, (2014).
11. Z. Bozdech *et al.*, The Transcriptome of the Intraerythrocytic Developmental Cycle of *Plasmodium falciparum*. *PLOS Biology* **1**, e5 (2003).
12. H. J. Painter *et al.*, Genome-wide real-time in vivo transcriptional dynamics during *Plasmodium falciparum* blood-stage development. *Nature Communications* **9**, 2656 (2018).
13. N. Ponts *et al.*, Nucleosome landscape and control of transcription in the human malaria parasite. *Genome Res* **20**, 228-238 (2010).
14. M. J. López-Barragán *et al.*, Directional gene expression and antisense transcripts in sexual and asexual stages of *Plasmodium falciparum*. *BMC Genomics* **12**, 587 (2011).

Reviewer #2 (Remarks to the Author):

The manuscript by Kioko et al has improved substantially, with important changes and additions that make the data more robust and the presentation of the results more accurate. The authors have addressed the majority of my comments, including toning down statements about conclusions that are speculative.

However, I still have a small number of additional suggestions, which can be addressed without performing new experiments or new analyses.

Lines 125-6. This sentence does not seem to be justified at this point of the manuscript. The phase shift described in this section of the results may be explained by multiple mechanisms and does not directly suggest this function for EVs.

I suggest to include the very clear schematics in Sup. Fig. 3 as main figures. I leave it up to the authors, but please note that 'delta mesor' is not a familiar concept for most malariologists.

Line 168. Why are *gdv1* sense transcripts classified as "unwanted". Why should the parasite 'not want' these important transcripts? Please revise.

Line 196. The statement "regulation of gene expression is believed to predominantly occur post-transcriptionally" is not correct. Notwithstanding the importance of posttranscriptional mechanisms in malaria, there is general agreement that the main mechanism of gene expression regulation in *P. falciparum* occurs at the transcriptional level, involving transcription factors of the ApiAP2 family. I am surprised that ApiAP2s are not even mentioned in the full manuscript. In a paper focused on periodicity of gene expression of malaria parasites, it is essential to discuss the central role of ApiAP2 transcription factors, both in the Introduction and in the Discussion.

Discussion. The hypothetical model proposed by the authors would imply a sophisticated mechanism to selectively include some specific RNAs in the EVs and exclude others. How could the RNAs that need to be eliminated be specifically recognized by RBPs or targeted by posttranscriptional modifications? This could be discussed, if the authors have a plausible hypothesis.

Lines 229-30. This sentence is not justified. No data is presented suggesting that the proposed RNA secretory system may be able to respond to threats or other conditions of the environment.

Line 246. Please provide a reference for the Dd2 line.

Ref. 42 and 47 are the same.

Sup. Fig. 5 is useful to rule out the possibility that all genes in clusters c1 and c2 are expressed at near background levels. However, I am intrigued by the different distribution of mean expression in WPs and EVs. While in WP it is apparent that c1 genes are lowly expressed genes and c6 are highly expressed genes, it seems as genes in all clusters are expressed at similar levels in EVs. This should be discussed and this informative figure moved to the main manuscript. It is very important for the interpretation of the whole dataset.

Reviewer #3 (Remarks to the Author):

The use of albumax indeed reduces the contamination with lipoproteins but does not exclude completely their presence. This is why, a deeper characterization of Pf-EVs was requested. Bead-

based FACS analysis with classical exosomal markers of samples used in this study, as performed with plasma-derived EVs as controls, would nicely demonstrate the absence of such putative "minor" contaminants. Moreover, glycophorin can be used as an excellent marker for Pf-EVs from in vitro culture (Neveu et al., 2020 Blood). Last, the EM image presented in Figure 1b could add a histogram of size distribution and an immuno-EM image against MSP1 or SBP1 to unequivocally show association with EVs.

Translational repression has been unarguably documented in gametocyte culture of *P. falciparum* (Lasonder et al., 2016 Nucleic Acids Res). This work presented unequivocal evidence of the existence of a large set of female gametocytes genes that are transcribed but not translated. Such transcripts, in principle, cannot be discarded in EVs during the IDC as they are essential for transmission of malaria parasites. This is why an experiment on gametocyte induction and analysis of TR genes in Pf-EVs was suggested. Of note, the data that transcription and translation are tightly coupled during the IDC, was not observed in a later study where it was found that there is a time-shift between the detection of the transcript and its cognate protein in specific gene families during the IDC (LeRoch et al., 2015 Genome Res).

Translation of EVs has been previously shown (Lai et al., 2014 Nat Comm) but this step is not an absolute requirement to observe EV-associated parasite proteins and mRNA in the same preparations. It is interesting to know that CSP has not been detected in the proteomics data generated by these authors.

The new discussion on alternative mechanisms adds value to the manuscript. Yet, it is unfortunate that the authors refuse to generate any functional evidence on this novel hypothesis. As such, it remains an interesting observation based on this descriptive dataset. Nature Communications have published several papers on EVs and malaria and in all of them functional assays to demonstrate novel hypothesis and/or advance mechanistic insights have been presented. Thus, in its present form this manuscript cannot be accepted for publication.

Reviewer 1

Lines 125-6. This sentence does not seem to be justified at this point of the manuscript. The phase shift described in this section of the results may be explained by multiple mechanisms and does not directly suggest this function for EVs.

We have deleted this sentence from the new version and replaced it with "The anticorrelation of large numbers of genes findings suggest that the secretion of RNA via *Pf*EVs is not random" (lines 143-144)

I suggest to include the very clear schematics in Sup. Fig. 3 as main figures. I leave it up to the authors, but please note that 'delta mesor' is not a familiar concept for most malariologists.

We agree that providing as much clarity on technical terms for all readers is essential. We have moved the schematics in Sup. Fig.3 to the main text as Fig.2 a-b as requested.

Line 168. Why are *gdv1* sense transcripts classified as "unwanted". Why should the parasite 'not want' these important transcripts? Please revise.

We refer to transcripts not required during the IDC as "unwanted" which applies to *GDV1* as it has no known function in asexual growth. However, we have revised this sentence by deleting the word "unwanted", which we have realised may be misinterpreted by readers. We also made a few other changes including the title of the section (lines 145, 165, 186-189, 55). ..

Line 196. The statement "regulation of gene expression is believed to predominantly occur post-transcriptionally" is not correct. Notwithstanding the importance of post-transcriptional mechanisms in malaria, there is general agreement that the main mechanism of gene expression regulation in *P. falciparum* occurs at the transcriptional level, involving transcription factors of the *ApiAP2* family. I am surprised that *ApiAP2*s are not even mentioned in the full manuscript. In a paper focused on periodicity of gene expression of malaria parasites, it is essential to discuss the central role of *ApiAP2* transcription factors, both in the introduction and in the discussion.

We agree, we did not previously accurately communicate the body of literature that shows an important role for transcriptional regulation, and have now corrected this by in the Introduction (lines 40-51) and in the Discussion (line 238).

Discussion. The hypothetical model proposed by the authors would imply a sophisticated mechanism to selectively include some specific RNAs in the EVs and exclude others. How could the RNAs that need to be eliminated be specifically recognised by RBPs or targeted by post-transcriptional modifications? This could be discussed, if the authors have a plausible hypothesis.

We hypothesise that post-transcriptional RNA modification (epitranscriptome marks) which has been reported in *P.falciparum* (Baumgarten, S. *et al*, *Nature microbiology*, doi:10.1038/s41564-019-0521-7 (2019)) may mark the secretion destined RNA for recognition by RNA binding proteins, which could then sequester the bound RNA into EVs as has been described previously for other systems (1, 2). This hypothesis is clearly laid out in the Discussion as suggested (lines 246-262).

Lines 229-30. This sentence is not justified. No data is presented suggesting that the proposed RNA secretory system may be able to respond to threats or other conditions of the environment.

We have deleted the above sentence.

Line 246. Please provide a reference for the Dd2 line.

We have now provided a reference for Dd2 as requested (line 79, 277)

Ref. 42 and 47 are the same.

We have removed the second duplicate of Painter et.al., and thank the reviewer for pointing this error out.

Sup. Fig. 5 is useful to rule out the possibility that all genes in clusters c1 and c2 are expressed at near background levels. However, I am intrigued by the different distribution of mean expression in WPs and EVs. While in WP it is apparent that c1 genes are lowly expressed genes and c6 are highly expressed genes, it seems as genes in all clusters are expressed at similar levels in EVs. This should be discussed and this informative figure moved to the main manuscript. It is very important for the interpretation of the whole dataset.

We agree with the reviewer that there appears to be a standard threshold of RNA secretion via PfEVs, and only genes that are transcribed above this threshold are physiologically required by the parasite during the IDC. Supplementary Fig 5 has also been moved to the main text (as Fig. 3b-c) to increase attention to this important point and we attempted to discuss it in lines 236-240.

Reviewer 3

The use of albumax indeed reduces the contamination with lipoproteins but does not exclude completely their presence. This is why, a deeper characterisation of Pf-EVs was requested. Bead-based FACS analysis with classical exosomal markers of samples used in this study, as performed with plasma-derived EVs as controls, would nicely demonstrate the absence of such putative "minor" contaminants. Moreover, glycophorin can be used as an excellent marker for Pf-EVs from in vitro culture (Neveu et al., 2020 Blood).

We appreciate the reviewer's recommendation here as the experiment produced a result that is helpful to our work in general. As requested, we performed a bead-based analysis of *P. falciparum* (Pf) medium EVs (formerly called microvesicles), Pf small EVs (formerly called exosome-like and were the focus of our study) and plasma-derived small EVs (formerly called exosomes) using classic exosome markers, CD63 and CD9 (Fig 1d, **Sup. Fig. 2e-g**). We found that Pf-EVs (both small and medium) contain relatively lower levels of CD63 and CD9 than plasma-derived small EVs (fig.1d). This suggests that our Pf small EVs from which we isolated RNA have appropriately low albumax-derived EVs contamination.

We also repeated the analysis using anti-glycophorin A as performed by Neveu et al. (3). We found that Pf small EVs (<200 nm diameter) do not contain GYPA on their surface. This is expected because rather than being membrane blebs, small EVs form are known to be generated as part of an active process (endosomal or non-degradative autophagy). The data we generated suggest that Pf small EVs originate from inside the infected erythrocytes and perhaps even from inside the parasite and are transported via three membranes to the extracellular milieu. However, GYPA expression was prominent in Pf medium EVs, which presumably form from RBCs membrane blebbing, and their diameter can be as large as 1000 nm. We checked the Neveu et.al paper suggested by the reviewer (3) and found that the authors adopted a protocol that recommends an 800 nm filter (4) instead of the 200 nm we used to isolate Pf small EVs. This implies that Neveu et.al. was enriching for Pf medium EVs (microvesicles) and explains why they detected GYPA in their Pf-EV pellets, although we cannot rule out

differences in reactivity of the anti-GTPA used. We have captured this information in lines 84-95 and 207-220 of the text and figure 1d and suppl figure 2.

Last, the EM image presented in Figure 1b could add a histogram of size distribution and an immuno-EM image against MSP1 or SBP1 to unequivocally show association with EVs.

As requested, we have provided a histogram of size distribution from EM images (**Fig 1c**). The focus of this study is on RNA but we hope to provide immuno-EM evidence in a similar study focusing on parasite proteins loaded into small PfEVs.

Translational repression has been unarguably documented in gametocyte culture of *P. falciparum* (Lasonder et al., 2016 Nucleic Acids Res). This work presented unequivocal evidence of the existence of a large set of female gametocytes genes that are transcribed but not translated. Such transcripts, in principle, cannot be discarded in EVs during the IDC as they are essential for transmission of malaria parasites. This is why an experiment on gametocyte induction and analysis of TR genes in Pf-EVs was suggested.

The work presented in this manuscript is the final product of over five years of work and similar work focusing on the gametocyte stage is a huge undertaking and deserves a dedicated study.

Besides, the Lasonder et al. 2016 study identified female and male gametocyte-specific transcripts in gametocyte cultures (5). The IDC stages (the focus of our study) do not require the identified female and male gametocyte-specific transcripts and may not need to store either, but gametocytes need to store these transcripts which are subsequently translated into proteins once the parasite is inside the mosquito (gametes, zygotes and ookinetes stages). They include Pf47 (required for mosquito immune evasion (6)), P230p (required for ookinete formation (7)) and CCps (1-4), which encode proteins required to form the crystalloid organelle in ookinetes (8). Therefore, we can only say that such gametocyte-specific transcripts, which mosquito stages require, cannot be discarded in EVs during gametocyte stages. However, we do not have such data because our current study only focused on the IDC stage. As far as we are aware, there is no current evidence that these gametocyte-specific transcripts identified by Lasonder et al. accumulate during the IDC; therefore, they presumably can be discarded via EVs during this stage without affecting subsequent transmission.

Of note, the data that transcription and translation are tightly coupled during the IDC, was not observed in a later study where it was found that there is a time-shift between the detection of the transcript and its cognate protein in specific gene families during the IDC (LeRoch et al., 2015 Genome Res).

The only study by LeRoch et al. which we could find that compared mRNA and protein levels in *P. falciparum* (9) was published in Genome Research in 2004 and not 2015, as indicated by the reviewer; apologies if we are missing the specific reference that they mean. In that 2004 study, the authors found that the protein levels from only 171 genes had a forward time-shift compared to the mRNA abundance, while 72 genes had a reverse time-shift. Furthermore, the authors also calculated sample-wise Spearman rank correlations between mRNA and protein levels and found rho values of >0.5 in merozoites, rings, trophozoites and schizonts.

By contrast, a more recent study by Caro et al., (10), which we referred to in our response to the first review, reported that transcription and translation were tightly coupled for >90% of the *P. falciparum* genes. Thus, both LeRoch et al. 2004 and Caro et al. 2014, suggest that only less than 10% (<500 genes) of *P. falciparum* genes might have protein levels shifted relative to mRNA levels. We believe that such low numbers compared to the entire transcriptome of >5000 transcripts make it more secure to say that for the majority of genes, transcription and translation are tightly coupled in *P. falciparum*, although a few exceptions have been suggested (LeRoch et.al., 2004).

Translation of EVs has been previously shown (Lai et al., 2014 Nat Comm), but this step is not an absolute requirement to observe EV-associated parasite proteins and mRNA in the same preparations. It is interesting to know that CSP has not been detected in the proteomics data generated by these authors.

We agree with the reviewer that translation in EVs is not an absolute requirement to observe EV-associated proteins and mRNA in the same preparations.

The new discussion on alternative mechanisms adds value to the manuscript. Yet, it is unfortunate that the authors refuse to generate any functional evidence on this novel hypothesis. As such, it remains an interesting observation based on this descriptive dataset. Nature Communications have published several papers on EVs and malaria and in all of them functional assays to demonstrate novel hypothesis

and/or advance mechanistic insights have been presented. Thus, in its present form this manuscript cannot be accepted for publication.

We appreciate the reviewer's enthusiasm in seeing functional assays that further strengthen the proposed alternative mechanism of RNA regulation deduced from our observational data. A strong observational evidence, obtained in an unbiased way, is provided in the manuscript to propose an alternative mechanism of post-transcriptional modulation of RNA in *P. falciparum*, which substantially contributes advances to the field that is worth sharing via NatureComms. While functional data is important, it requires access to more powerful tools than those currently available to us and it constitutes an extensive piece of work in its own right. We do hope such studies will be performed by us or others in the future in follow-up work.

References

1. A. M. Leidal *et al.*, The LC3-conjugation machinery specifies the loading of RNA-binding proteins into extracellular vesicles. *Nature cell biology* **22**, 187-199 (2020).
2. A. M. Leidal, J. Debnath, LC3-dependent extracellular vesicle loading and secretion (LDELS). *Autophagy* **16**, 1162-1163 (2020).
3. G. Neveu *et al.*, Plasmodium falciparum sexual parasites develop in human erythroblasts and affect erythropoiesis. *Blood* **136**, 1381-1393 (2020).
4. S. Mbagwu, M. Walch, L. Filgueira, P. Y. Mantel, Production and Characterization of Extracellular Vesicles in Malaria. *Methods Mol Biol* **1660**, 377-388 (2017).
5. E. Lasonder *et al.*, Integrated transcriptomic and proteomic analyses of *P. falciparum* gametocytes: molecular insight into sex-specific processes and translational repression. *Nucleic Acids Res* **44**, 6087-6101 (2016).
6. A. Molina-Cruz, G. E. Canepa, C. Barillas-Mury, Plasmodium P47: a key gene for malaria transmission by mosquito vectors. *Curr Opin Microbiol* **40**, 168-174 (2017).
7. C. Marin-Mogollon *et al.*, The Plasmodium falciparum male gametocyte protein P230p, a paralog of P230, is vital for ookinete formation and mosquito transmission. *Scientific Reports* **8**, 14902 (2018).
8. R. Jenwithisuk *et al.*, Identification of a PH domain-containing protein which is localized to crystalloid bodies of Plasmodium ookinetes. *Malaria Journal* **17**, 466 (2018).
9. K. G. Le Roch *et al.*, Global analysis of transcript and protein levels across the Plasmodium falciparum life cycle. *Genome Res* **14**, 2308-2318 (2004).
10. F. Caro, V. Ahyong, M. Betegon, J. L. DeRisi, Genome-wide regulatory dynamics of translation in the Plasmodium falciparum asexual blood stages. *Elife* **3**, (2014).

Reviewer comments, third round

Reviewer #3 (Remarks to the Author):

I appreciate the new reply to my queries. In spite of functional evidence, this manuscript is now ready for publication